# Parameter Expanded Stochastic Gradient Markov Chain Monte Carlo

**Hyunsu Kim, Giung Nam, Chulhee Yun, Hongseok Yang, & Juho Lee**
KAIST, South Korea
{kim.hyunsu,giung,chulhee.yun,hongseok.yang,juholee}@kaist.ac.kr

## Abstract

Bayesian Neural Networks (BNNs) provide a promising framework for modeling predictive uncertainty and enhancing out-of-distribution robustness (OOD) by estimating the posterior distribution of network parameters. Stochastic Gradient Markov Chain Monte Carlo (SGMCMC) is one of the most powerful methods for scalable posterior sampling in BNNs, achieving efficiency by combining stochastic gradient descent with second-order Langevin dynamics. However, SGMCMC often suffers from limited sample diversity in practice, which affects uncertainty estimation and model performance. We propose a simple yet effective approach to enhance sample diversity in SGMCMC without the need for tempering or running multiple chains. Our approach reparameterizes the neural network by decomposing each of its weight matrices into a product of matrices, resulting in a sampling trajectory that better explores the target parameter space. This approach produces a more diverse set of samples, allowing faster mixing within the same computational budget. Notably, our sampler achieves these improvements without increasing the inference cost compared to the standard SGMCMC. Extensive experiments on image classification tasks, including OOD robustness, diversity, loss surface analyses, and a comparative study with Hamiltonian Monte Carlo, demonstrate the superiority of the proposed approach.

## 1 Introduction

Bayesian Neural Networks (BNNs) provide a promising framework for achieving predictive uncertainty and out-of-distribution (OOD) robustness (Goan & Fookes, 2020). Instead of using a gradient-descent optimization algorithm typical for neural networks (NNs), BNNs are trained by estimating the posterior distribution $p(\theta|\mathcal{D})$, where $\theta$ represents the BNN parameters and $\mathcal{D}$ is the given dataset. Variational methods (Liu & Wang, 2016; David M. Blei & McAuliffe, 2017) and sampling methods are commonly used to estimate the posterior. A representative sampling method is Stochastic Gradient Markov Chain Monte Carlo (SGMCMC; Welling & Teh, 2011; Chen et al., 2014; Ma et al., 2015), which has been highly successful for large-scale Bayesian inference by leveraging both the exploitation of stochastic gradient descent (SGD) and the exploration of Hamiltonian Monte Carlo (HMC; Duane et al., 1987). Based on an appropriate stochastic differential equation, SGMCMC approximately draws samples from the posterior, while avoiding the computation of the intractable posterior density $p(\theta|\mathcal{D})$ and instead using its unnormalized tractable counterpart $p(\mathcal{D}|\theta)p(\theta)$.

However, despite its powerful performance, SGMCMC suffers from the issue of low sample diversity. Drawing diverse samples of the BNN parameters $\theta$ is important because it usually leads to the diversity of the sampled functions $f_\theta$ of the BNN, which in turn induces an improved approximation of the likelihood $p(\mathcal{D}|\theta)$. As a result, SGMCMC often fails to replace less-principled alternatives in practice, in particular, deep ensemble (DE; Lakshminarayanan et al., 2017), which generates multiple samples of the BNN parameters $\theta$ simply by training a neural network multiple times with different random initializations and SGD. Although DE is technically not a principled Bayesian method, it partially approximates the posterior with high sample diversity, leading to a good uncertainty estimation (Ovadia et al., 2019; Ashukha et al., 2020). When training time is not a concern, DE usually outperforms SGMCMC in terms of both accuracy and uncertainty estimation.

Existing approaches for addressing the sample-diversity issue of SGMCMC mostly focus on modifying the dynamics of SGMCMC directly, e.g., by adjusting the step size schedule (Zhang et al., 2020), introducing preconditioning into the dynamics (Ma et al., 2015; Gong et al., 2019; Kim et al., 2024), or running multiple SGMCMC chains to explore different regions of the loss surface (Gallego & Insua, 2020; Deng et al., 2020). However, such a direct modification of the dynamics commonly requires extra approximations, such as the estimation of the Fisher information, bi-level optimization to learn appropriate preconditioning, or high computational resources, such as a large amount of memory due to the use of multiple chains.

In this paper, we propose a simple yet effective approach for increasing the sample diversity of SGMCMC without requiring explicit preconditioning or multiple chains. Our approach modifies the dynamics of SGMCMC *indirectly* by reparameterizing the BNN parameters. In the approach, the original parameter matrices of the BNN are decomposed into the products of matrices of new parameters. Specifically, for a given multilayer perceptron (MLP), when $\mathbf{W} \in \mathbb{R}^{m \times n}$ is a parameter matrix of an MLP layer, our approach reparameterizes $\mathbf{W}$ as the following matrix product:

$$\mathbf{W} = \mathbf{PVQ}, \tag{1}$$

for new parameter matrices $\mathbf{V} \in \mathbb{R}^{m \times n}$, $\mathbf{P} \in \mathbb{R}^{m \times m}$ and $\mathbf{Q} \in \mathbb{R}^{n \times n}$ for the same layer. We call the approach as *Parameter Expanded SGMCMC (PX-SGMCMC)*. In the paper, we provide theoretical and empirical evidence that our reparametrization alters the dynamics of SGMCMC such that PX-SGMCMC explores the target potential energy surface better than the original SGMCMC. The modified dynamics introduce a preconding on the gradient of the potential energy and causes an effect of increasing step size implicitly. While simply growing the step size often hinders the convergence of gradient updates, the implicit step size scaling caused by the precondintioning improves the convergence. Although PX-SGMCMC needs more BNN parameters during training, its inference cost remains the same as SGMCMC because the matrices $\mathbf{P}, \mathbf{V}, \mathbf{Q}$ in Eq. 1 can be reassembled into the single weight matrix $\mathbf{W}$ for inference.

We evaluate the performance of PX-SGMCMC on various image classification tasks, with residual networks (He et al., 2016), measuring both in-distribution and OOD performance. Furthermore, we assess sample diversity in various ways, such as measuring ensemble ambiguity, comparing PX-SGMCMC with HMC (which is considered an oracle method), and visualizing the sampling trajectories over the loss surface. Our evaluation shows that PX-SGMCMC outperforms SGMCMC and other baselines by a significant margin, producing more diverse function samples and achieving better uncertainty estimation and OOD robustness than these baselines.

## 2 PRELIMINARIES

### 2.1 NOTATION

We begin by presenting the mathematical formulation of neural networks. Specifically, a multilayer perceptron (MLP; Rosenblatt, 1958) with $L$ layers transforms inputs $\boldsymbol{x} = \boldsymbol{h}^{(0)}$ to outputs $\boldsymbol{y} = \boldsymbol{h}^{(L)}$ through the following transformations:

$$\boldsymbol{h}^{(l)} = \sigma(\mathbf{W}^{(l)}\boldsymbol{h}^{(l-1)} + \boldsymbol{b}^{(l)}), \text{ for } l = 1, \ldots, L-1, \text{ and } \boldsymbol{h}^{(L)} = \mathbf{W}^{(L)}\boldsymbol{h}^{(L-1)} + \boldsymbol{b}^{(L)}, \tag{2}$$

where $\boldsymbol{h}^{(l)}$ denotes the feature at the $l$-th layer, $\mathbf{W}^{(l)}$ and $\boldsymbol{b}^{(l)}$ respectively are the weight and bias parameters at the $l$-th layer, and $\sigma(\cdot)$ indicates the activation function applied element-wise.

### 2.2 BAYESIAN INFERENCE WITH STOCHASTIC GRADIENT MCMC

**Bayesian model averaging.** In Bayesian inference, our goal is not to find a single best estimate of the parameters, such as the maximum a posteriori estimate, but instead to sample from the posterior distribution $p(\boldsymbol{\theta}|\mathcal{D})$ of the parameters $\boldsymbol{\theta}$ given the observed data $\mathcal{D}$. The prediction for a new datapoint $x$ is then given by Bayesian model averaging (BMA),

$$p(y|x, \mathcal{D}) = \int p(y|x, \boldsymbol{\theta}) p(\boldsymbol{\theta}|\mathcal{D}) \mathrm{d}\boldsymbol{\theta}, \tag{3}$$

which can be approximated by Monte Carlo integration $p(y|x, \mathcal{D}) \approx \sum_{m=1}^{M} p(y|x, \boldsymbol{\theta}_m)/M$ using finite posterior samples $\boldsymbol{\theta}_1, \ldots, \boldsymbol{\theta}_M \sim p(\boldsymbol{\theta}|\mathcal{D})$. This Monte-Carlo integration is commonly used

in Bayesian deep learning with posterior samples generated by a sampling method, as the posterior distribution of the neural network parameters $\boldsymbol{\theta}$ cannot be expressed in closed form in practice.

**Langevin dynamics for posterior simulation.** Due to the intractability of the posterior $p(\boldsymbol{\theta}|\mathcal{D})$ for the neural network parameters $\boldsymbol{\theta}$, we often work with its unnormalized form. Specifically, we typically introduce the *potential energy*, defined as the negative of the unnormalized log-posterior:

$$U(\boldsymbol{\theta}) = -\log p(\mathcal{D}|\boldsymbol{\theta}) - \log p(\boldsymbol{\theta}). \tag{4}$$

Simulating the following Langevin dynamics over the neural network parameters $\boldsymbol{\theta}$,

$$\mathrm{d}\boldsymbol{\theta} = \mathbf{M}^{-1}\boldsymbol{r}\mathrm{d}t, \quad \mathrm{d}\boldsymbol{r} = -\gamma\boldsymbol{r}\mathrm{d}t - \nabla_{\boldsymbol{\theta}}U(\boldsymbol{\theta})\mathrm{d}t + \mathbf{M}^{1/2}\sqrt{2\gamma T}\mathrm{d}\mathcal{W}, \tag{5}$$

yields a trajectory distributed according to $\exp\left(-U(\boldsymbol{\theta})/T\right)$, where setting $T = 1$ provides posterior samples for computing the BMA integral (Eq. 3). Here, $\mathbf{M}$ is the mass, $\gamma$ is the damping constant, $\mathcal{W}$ represents a standard Wiener process, and $T$ is the temperature.

**Langevin dynamics with stochastic gradients.** Computing the gradient $\nabla_{\boldsymbol{\theta}}U(\boldsymbol{\theta})$ over the entire dataset $\mathcal{D}$ becomes intractable as the dataset size increases. Inspired by stochastic gradient methods (Robbins & Monro, 1951), SGMCMC introduces a noisy estimate of the potential energy:

$$\tilde{U}(\boldsymbol{\theta}) = -(|\mathcal{D}|/|\mathcal{B}|)\log p(\mathcal{B}|\boldsymbol{\theta}) - \log p(\boldsymbol{\theta}), \tag{6}$$

where the log-likelihood is computed only for a mini-batch of data $\mathcal{B} \subset \mathcal{D}$, replacing the full-data gradient with the mini-batch gradient. In practice, simulations rely on the semi-implicit Euler method, with Stochastic Gradient Langevin Dynamics (SGLD; Welling & Teh, 2011) and Stochastic Gradient Hamiltonian Monte Carlo (SGHMC; Chen et al., 2014) being two representatives:

$$(\text{SGLD}) \quad \boldsymbol{\theta} \leftarrow \boldsymbol{\theta} - \epsilon\mathbf{M}^{-1}\nabla_{\boldsymbol{\theta}}\tilde{U}(\boldsymbol{\theta}) + \mathcal{N}(\mathbf{0}, 2\epsilon T\mathbf{M}), \tag{7}$$

$$(\text{SGHMC}) \quad \boldsymbol{\theta} \leftarrow \boldsymbol{\theta} + \epsilon\mathbf{M}^{-1}\boldsymbol{r}, \quad \boldsymbol{r} \leftarrow (1 - \epsilon\gamma)\boldsymbol{r} - \epsilon\nabla_{\boldsymbol{\theta}}\tilde{U}(\boldsymbol{\theta}) + \mathcal{N}(\mathbf{0}, 2\epsilon\gamma T\mathbf{M}). \tag{8}$$

## 3 PARAMETER EXPANDED SGMCMC

### 3.1 REPARAMETRIZATION

Our solution for the sample-diversity problem of SGMCMC is motivated by the intriguing prior results on deep linear neural networks (DLNNs) (Arora et al., 2018; 2019a; He et al., 2024), which are just MLPs with a linear or even the identity activation function (i.e., $\sigma(t) = t$). Although DLNNs are equivalent to linear models in terms of expressiveness, their training trajectories during (stochastic) gradient descent are very different from those of the corresponding linear models. Existing results show that, under proper assumptions, DLNNs exhibit faster convergence (Arora et al., 2018; 2019a) and have an implicit bias (distinct from linear models) to converge to solutions that generalize better (Woodworth et al., 2020; Arora et al., 2019b; Gunasekar et al., 2018). Observe also that all the layers of each DLNN can be reassembled into a single linear layer so that the DLNNs do not incur overhead during inference when compared to the linear models.

Building on these results and observation, we introduce the *expanded parametrization* (EP) of an $L$-layer MLP $f$ in Eq. 2 with parameters $\boldsymbol{\theta} = (\mathbf{W}^{(1)}, \ldots, \mathbf{W}^{(L)}, \boldsymbol{b}^{(1)}, \ldots, \boldsymbol{b}^{(L)})$ as

$$\mathbf{W}^{(l)} = \mathbf{P}_{1:c}^{(l)}\mathbf{V}^{(l)}\mathbf{Q}_{1:d}^{(l)} \quad \text{and} \quad \boldsymbol{b}^{(l)} = \mathbf{P}_{1:c}^{(l)}\boldsymbol{a}^{(l)}, \text{ for } l = 1, \ldots, L, \tag{9}$$

where $\mathbf{P}_{1:c}^{(l)} \triangleq \mathbf{P}_c^{(l)} \cdots \mathbf{P}_1^{(l)}$ and $\mathbf{Q}_{1:d}^{(l)} \triangleq \mathbf{Q}_1^{(l)} \cdots \mathbf{Q}_d^{(l)}$ for some new parameter matrices $\mathbf{P}_i^{(l)}$ and $\mathbf{Q}_j^{(l)}$, called *expanded matrices*. Here, $c$ represents the number of expanded matrices on the left side, and $d$ on the right side, and the total number of expanded matrices is $e = c + d \geq 0$ (note that when $c = d = 0$, $\mathbf{P}_{1:0}^{(l)}$ and $\mathbf{Q}_{1:0}^{(l)}$ are identity matrices). While the expanded matrices $\mathbf{P}_i^{(l)}$ and $\mathbf{Q}_i^{(l)}$ do not need to be square, their products, $\mathbf{P}_{1:c}^{(l)}$ and $\mathbf{Q}_{1:d}^{(l)}$, must be so in order to ensure that the *base matrix* $\mathbf{V}^{(l)}$ retains the dimensionality of $\mathbf{W}^{(l)}$. In this paper, we use square matrices for $\mathbf{P}_i^{(l)}$'s and $\mathbf{Q}_i^{(l)}$'s, which lets us minimize additional memory overhead while making sure that the reparametrization does not introduce any additional non-global local minima; this is guaranteed when the widths of intermediate layers in a DLNN are greater than or equal to both the input and output dimensions (Laurent & von Brecht, 2018; Yun et al., 2019).

Under our EP, the position variable in the SGLD algorithm is given by

$$\boldsymbol{\theta} = \left(\mathbf{P}_1^{(1)}, \ldots, \mathbf{P}_c^{(l)}, \mathbf{V}^{(l)}, \mathbf{Q}_1^{(1)}, \ldots, \mathbf{Q}_d^{(l)}, \boldsymbol{a}^{(l)}\right)_{l=1}^L, \tag{10}$$

while in the SGHMC algorithm, the momentum variable, in addition to the position, is defined as

$$\boldsymbol{r} = \left(\mathbf{R}_1^{(1)}, \ldots, \mathbf{R}_c^{(l)}, \mathbf{S}^{(l)}, \mathbf{T}_1^{(1)}, \ldots, \mathbf{T}_d^{(l)}, \boldsymbol{c}^{(l)}\right)_{l=1}^L, \tag{11}$$

where this new momentum variable is related to the original momentum variable of SGHMC as in Eq. 9. SP only has $\mathbf{S}$, while $\mathbf{R}$ and $\mathbf{T}$ are introduced by the expanded parameters in the EP setting. We call SGLD and SGHMC under these EPs broadly as *Parameter Expanded SGMCMC (PX-SGMCMC)* methods. Although the dynamics of such a PX-SGMCMC method follows the update formulas in Eq. 7, it differs from the dynamics of the corresponding SGMCMC method significantly. The former is the preconditioned variant of the latter where gradients in the SGMCMC's dynamics, such as $\nabla_{\boldsymbol{\theta}_{\text{orig}}} \tilde{U}(\boldsymbol{\theta}_{\text{orig}})$ for the original position variable $\boldsymbol{\theta}_{\text{orig}}$, are replaced by preconditioned versions, and this preconditioning produces extraordinary directions of gradient steps in the PX-SGMCMC. In the next section, we dive into the new dynamics induced by the preconditioning, and analyze the effect of the preconditioning on the exploration of PX-SGMCMC in terms of the depth of EP and the maximum singular values of matrices in it.

## 3.2 THEORETICAL ANALYSIS

In this section, we provide a theoretical analysis of the EP proposed in Section 3.1. For simplicity, the following theorem and proof focus on the SGLD method described in Eq. 7. To highlight the effect of EP on SGLD, we first need to understand how the combined parameter $\mathbf{W}$ evolves under gradient flows when each of its components follows its own gradient flow. The following lemma explains the preconditioning matrix induced by EP. Proofs can be found in Appendix A.

**Lemma 3.1** (Dynamics of EP). *For an arbitrary function $\mathcal{F}$ whose parameter is $\mathbf{W}_{1:e} = \mathbf{W}_1 \cdots \mathbf{W}_e$ with its vectorization $\mathbf{X} = \text{vec}(\mathbf{W}_{1:e})$, assume that the gradient update of each $\mathbf{W}_i$ for $i \in \{1, \ldots, e\}$ is defined as the following PDE:*

$$\mathrm{d}\mathbf{W}_i(t) = -\nabla_{\mathbf{W}_i} \mathcal{F}(\mathbf{W}_1(t), \ldots, \mathbf{W}_e(t))\mathrm{d}t. \tag{12}$$

*Then, their multiplication $\mathbf{X}$ satisfies the following dynamics:*

$$\mathrm{d}\mathbf{X}(t) = -P_{\mathbf{X}(t)} \nabla_{\mathbf{X}} \mathcal{F}(\mathbf{X}(t))\mathrm{d}t,$$

$$\text{where} \quad P_{\mathbf{X}(t)} = \begin{cases} I, & (e = 1), \\ \mathbf{W}_2(t)^\top \mathbf{W}_2(t) \otimes I + I \otimes \mathbf{W}_1(t)\mathbf{W}_1(t)^\top, & (e = 2) \\ \mathbf{W}_{2:e}(t)^\top \mathbf{W}_{2:e}(t) \otimes I + I \otimes \mathbf{W}_{1:e-1}(t)\mathbf{W}_{1:e-1}(t)^\top \\ \quad + \sum_{j=2}^{e-1} \left(\mathbf{W}_{j+1:e}(t)^\top \mathbf{W}_{j+1:e}(t) \otimes \mathbf{W}_{1:j-1}(t)\mathbf{W}_{1:j-1}(t)^\top\right) & (e > 2). \end{cases}$$

*The operator $\otimes$ refers to the Kronecker product, and $P_{\mathbf{X}(t)}$ denotes the symmetric and positive semi-definite matrix.*

This unique gradient flow is known to be unattainable through regularization (Arora et al., 2018) in the standard parameterization (SP). Furthermore, the singular values or eigenvalues of $P_{\mathbf{X}}(t)$ are closely tied to the singular values of the parameters $\mathbf{W}_i$. In the following, we show that the new dynamics induced by EP promotes greater exploration of the energy surface, which scales with the *depth* of EP and the *maximum singular value* across the EP parameters $P_i(t), V(t), Q_i(t)$ in Eq. 9.

**Theorem 3.2** (Exploration). *Assume the following bounds on the expectations of the norms of the gradients, the stochastic gradients, and the Gaussian noise in Eq. 7:*

$$\mathbb{E}\left[\left\|\nabla U(\mathbf{W}^{(l)}(t))\right\|_2\right] \le h, \quad \mathbb{E}\left[\left\|\nabla U(\mathbf{W}^{(l)}(t)) - \nabla \tilde{U}(\mathbf{W}^{(l)}(t))\right\|_2\right] \le s, \quad \mathbb{E}[\|2T\mathbf{M}\|_2] \le C, \tag{13}$$

*where the elements of $\mathbf{M}$ corresponding to the expanded parameters $\mathbf{P}, \mathbf{Q}$ are zero. Also assume that the maximum singular value of each parameter matrix in EP is bounded as follows:*

$$\sup_t \mathcal{V}(t) = M,$$

$$\mathcal{V}(t) = \max\left\{\sigma_{max}(\mathbf{P}_1(t)), \ldots, \sigma_{max}(\mathbf{P}_c(t)), \sigma_{max}(\mathbf{V}(t)), \sigma_{max}(\mathbf{Q}_1(t)), \ldots, \sigma_{max}(\mathbf{Q}_d(t))\right\}. \tag{14}$$

*Then, due to the preconditioning in Lemma 3.1, the Euclidean distance of two SGLD samples at consecutive time steps is upper-bounded by the following term, which depends on the depth $c+d+1$:*

$$\mathbb{E}\left[\|\mathbf{W}(t) - \mathbf{W}(t+1)\|_2\right] \leq \epsilon L^2(c+d+1)M^{(c+d)}(h+s) + \epsilon LC. \tag{15}$$

*Note that as the **depth** (c+d+1) and the **maximum singular value** M decrease, the upper bound on the distance of two consecutive samples gets smaller.*

Although the bound in Theorem 3.2 is not necessarily the maximum distance between consecutive samples, its dependency on the depth $(c+d+1)$ of our EP suggests that the preconditioning induced by EP may improve the exploration of the SGLD and lead to the generation of more diverse samples. Note that the bound in Eq. 15 is also proportional to the step size $\epsilon$. However, in practice, using a large step size above a certain threshold rather *hinders* the performance. In our experiments, the performance monotonically improved as we increased the depth of EP, while converging to a certain fixed level in the end. This suggests that the preconditioning of our EP induces a form of *implicit step-size scaling* while maintaining the stability of the gradient-descent steps.

### 3.3 Implementing EP for Different Architectures

**Linear layers.** We follow the Eq. 9. The depth and width of each matrix $\mathbf{P}_i$, $\mathbf{V}$, or $\mathbf{Q}_i$ can be adjusted, but the width should be at least as large as that of the corresponding input dimension.

**Convolution layers.** There are four dimension axes in the standard convolution layer. Let $k, c_i, c_o$ be the sizes of the kernel, the input channel, and the output channel, and $\mathbf{W} \in \mathbb{R}^{k \times k \times c_o \times c_i}$ be the kernel matrix. If we naïvely reparameterize $\mathbf{W}$ with $\mathbf{P} \in \mathbb{R}^{kc_o \times kc_o}$, $\mathbf{V} \in \mathbb{R}^{k \times k \times c_o \times c_i}$, and $\mathbf{Q} \in \mathbb{R}^{kc_i \times kc_i}$, the memory and computation overheads become significant. Thus, we let $\mathbf{P}$ and $\mathbf{Q}$ operate on the channel $c_i, c_o$ dimensions only, and each kernel dimension is multiplied by the same matrices by defining $\mathbf{P} \in \mathbb{R}^{c_o \times c_o}$, $\mathbf{V} \in \mathbb{R}^{k \times k \times c_o \times c_i}$, and $\mathbf{Q} \in \mathbb{R}^{c_i \times c_i}$. That is, in the index notation, our reparameterization is:

$$W_{abij} = \sum_{u,l} P_{iu}V_{abul}Q_{lj}, \quad b_i = \sum_u P_{iu}a_u. \tag{16}$$

In contrast to the naïve reparametrization, where the number of parameters roughly increases by three folds when the depth of the reparametrization is 3 (i.e., #Params($\mathbf{PVQ}$) = $3 \cdot$ #Params($\mathbf{W}$)), the above reparametrization requires only #Params($\mathbf{PVQ}$) = $(1+2/k^2) \cdot$ #Params($\mathbf{W}$) in that case.

**Normalization layers.** Normalization layers, such as Batch Normalization (Ioffe & Szegedy, 2015), Layer Normalization (Ba, 2016), and Filter Response Normalization (FRN; Singh & Krishnan, 2020), contain a few parameter vectors. For example, FRN used in Izmailov et al. (2021) consists of the scale, bias, and threshold vectors, $\boldsymbol{s} \in \mathbb{R}^{c_o}$, $\boldsymbol{b} \in \mathbb{R}^{c_o}$, $\boldsymbol{t} \in \mathbb{R}^{c_o}$. As in the case of the linear layer, we can simply multiply matrices on the left-hand side.

$$s_i = \sum_{u,l} P_{iu}^s Q_{ul}^s s_l, \quad b_i = \sum_{u,l} P_{iu}^b Q_{ul}^b b_l, \quad t_i = \sum_{u,l} P_{iu}^t Q_{ul}^t t_l. \tag{17}$$

The row and column dimensions of $P$ and $Q$ should be at least $c_o$.

## 4 Related Work

**Linear parameter expansion.** Linear parameter expansion techniques are relatively underexplored in deep learning, as they do not inherently increase the expressivity of deep neural networks, particularly in non-linear architectures. While this approach has shown some benefits in linear networks, it is often overlooked in modern deep learning applications. A few notable works, however, have employed parameter expansion in the context of convolutional neural networks. For instance, Chollet (2017), Guo et al. (2020), and Cao et al. (2022) have introduced techniques that either decompose or augment convolution layers to reduce FLOPs and improve generalization. These methods primarily aim at enhancing efficiency or regularization, leveraging additional layers or decompositions to modify the structure of network without significantly increasing computational complexity. On the other hand, Ding et al. (2019) proposed a different approach by expanding convolutional layers

through addition rather than multiplication, aiming to improve robustness against rotational distortions in input images. While their method enhances robustness to certain image transformations, it does not focus on the exploration properties of parameter expansion in non-convex problems.

**SGMCMCs for diversity.** Building upon the seminal work of Welling & Teh (2011), which introduced SGLD as a scalable MCMC algorithm based on stochastic gradient methods (Robbins & Monro, 1951), a range of SGMCMC methods have emerged in the past decade (Ahn et al., 2012; Patterson & Teh, 2013; Chen et al., 2014; Ding et al., 2014; Ma et al., 2015; Li et al., 2016). Despite their theoretical convergence to target posteriors under the Robbins–Monro condition with a decaying step size (Teh et al., 2016; Chen et al., 2015), effectively exploring and exploiting the posterior density of deep neural networks using a single MCMC chain remains challenging due to their multimodal nature. In this context, Zhang et al. (2020) proposed a simple yet effective cyclical step size schedule to enhance the exploration of SGMCMC methods. Intuitively, the larger step size at the beginning of each sampling cycle facilitates better exploration while tolerating simulation error, whereas the smaller step size towards the end of the cycle ensures more accurate simulation. While the cyclical schedule helps with exploration, it often struggles to fully capture multimodality and typically requires a significant number of update steps to transition between modes (Fort et al., 2019; Kim et al., 2024). Thus, improving SGMCMC methods for modern deep neural networks is still an active area of research, with recent progress using meta-learning frameworks (Gong et al., 2019; Kim et al., 2024). To the best of our knowledge, we are the first to propose the parameter expansion for enhancing the exploration of SGMCMC.

## 5 EXPERIMENTS

In this section, we present empirical results demonstrating the effectiveness of the parameter expansion strategy proposed in Section 3 for image classification tasks. We aim to show that PX-SGHMC, an SGHMC variant of PXSGMCMC, collects diverse posterior samples and consistently outperforms baseline methods across various tasks. We compare with the following SGMCMC methods as baselines: SGLD (Welling & Teh, 2011), pSGLD (Li et al., 2016), SGHMC (Chen et al., 2014), and SGNHT (Ding et al., 2014). Unless otherwise specified, all methods utilize a cyclical step size schedule. Note that in our method, we set the elements of the semidefinite damping (friction) $\gamma$ in Eq. 8 corresponding to **P** and **Q** much smaller than those of **V**, ensuring a stable SGHMC trajectory. For details on implementations and hyperparameters, please refer to Appendices C and D.

We conduct experiments including multimodal distribution (Section 5.1) sampling and Bayesian neural networks on image classification tasks (Sections 5.2 and 5.3). Unless stated otherwise, (1) the reported values are presented as "mean±std," averaged over four trials, and (2) a **bold-faced underline** highlights the best outcome, while an underline indicates the second-best value. To assess the quality of classifier predictions, we compute classification error (ERR) and negative log-likelihood (NLL), while ensemble ambiguity (AMB) quantifies the diversity of ensemble predictions. Refer to Appendix B.1 for the definitions of the evaluation metrics.

### 5.1 TOY RESULTS: MIXTURE OF GAUSSIANS

We begin by comparing SP and EP on the multimodal 2D mixture of 25 Gaussians (MoG), following Zhang et al. (2020). Although this distribution is not a BNN posterior, it is sufficient to demonstrate the role of the preconditioning induced by EP. The random variables of MoG, $\mathbf{x} \in \mathbb{R}^2$, are decomposed as $\mathbf{W}_3 \mathbf{W}_2 \mathbf{W}_1 \mathbf{x}$ for $\mathbf{W}_1, \mathbf{W}_2, \mathbf{W}_3 \in \mathbb{R}^{2 \times 2}$, and HMC sampling also follows the preconditioning on the gradients. Specifically, we collect 10,000 samples by running HMC with a single chain, using 10 leapfrog steps and a constant step size of 0.05 for both SP and EP. Fig. 1 illustrates that EP resolves the issue of SP getting trapped in local modes

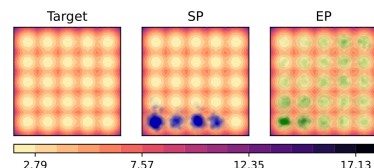

**Figure 1: Toy results.** HMC samples with SP and EP. The color represents the negative log probability.

without requiring additional techniques such as tempering or cyclical step sizes (Zhang et al., 2020). As explained in Section 3.2, the implicit step size scaling induced by the preconditioning facilitates escaping local modes, which suggests strong performance in non-convex problems such as multimodal distributions.

**Table 1: Main results on CIFAR-10 and associated distribution shifts.** Evaluation results on C10 (CIFAR-10), C10.1 (CIFAR-10.1), C10.2 (CIFAR-10.2), STL, and C10-C (CIFAR-10-C). Metrics for C10-C are computed using a total of 950,000 examples, encompassing 5 intensity levels and 19 corruption types. Please refer to Appendix B.2 for more detailed results regarding C10-C.

| Metric | Method | C10 | Distribution shifts | | | |
|---|---|---|---|---|---|---|
| | | | C10.1 | C10.2 | STL | C10-C |
| ERR (↓) | SGLD | $0.152_{\pm0.003}$ | $0.258_{\pm0.007}$ | $0.293_{\pm0.004}$ | $0.326_{\pm0.004}$ | $0.309_{\pm0.007}$ |
| | pSGLD | $0.158_{\pm0.004}$ | $0.277_{\pm0.004}$ | $0.305_{\pm0.003}$ | $0.333_{\pm0.003}$ | $0.316_{\pm0.006}$ |
| | SGHMC | $\underline{0.133}_{\pm0.002}$ | $\underline{0.234}_{\pm0.003}$ | $0.277_{\pm0.006}$ | $\underline{0.306}_{\pm0.003}$ | $\underline{0.292}_{\pm0.002}$ |
| | SGNHT | $0.135_{\pm0.002}$ | $0.236_{\pm0.005}$ | $\underline{0.273}_{\pm0.004}$ | $0.307_{\pm0.005}$ | $0.294_{\pm0.008}$ |
| | **PX-SGHMC (ours)** | $\mathbf{0.121}_{\pm0.002}$ | $\mathbf{0.218}_{\pm0.005}$ | $\mathbf{0.257}_{\pm0.007}$ | $\mathbf{0.287}_{\pm0.002}$ | $\mathbf{0.287}_{\pm0.008}$ |
| NLL (↓) | SGLD | $0.477_{\pm0.009}$ | $0.772_{\pm0.010}$ | $0.891_{\pm0.005}$ | $0.928_{\pm0.014}$ | $0.913_{\pm0.022}$ |
| | pSGLD | $0.501_{\pm0.007}$ | $0.812_{\pm0.015}$ | $0.928_{\pm0.007}$ | $0.958_{\pm0.006}$ | $0.938_{\pm0.017}$ |
| | SGHMC | $\underline{0.422}_{\pm0.005}$ | $\underline{0.698}_{\pm0.006}$ | $0.834_{\pm0.007}$ | $\underline{0.868}_{\pm0.010}$ | $\underline{0.871}_{\pm0.005}$ |
| | SGNHT | $0.425_{\pm0.004}$ | $0.705_{\pm0.007}$ | $\underline{0.833}_{\pm0.008}$ | $0.873_{\pm0.006}$ | $0.877_{\pm0.021}$ |
| | **PX-SGHMC (ours)** | $\mathbf{0.388}_{\pm0.005}$ | $\mathbf{0.661}_{\pm0.012}$ | $\mathbf{0.806}_{\pm0.009}$ | $\mathbf{0.819}_{\pm0.004}$ | $\mathbf{0.859}_{\pm0.022}$ |

**Table 2: Out-of-distribution detection.** Evaluation results for distinguishing in-distribution inputs from CIFAR-10 and out-of-distribution inputs from SVHN and LSUN based on predictive entropy. This table summarizes evaluation metrics, including AUROC, TNR95 (TNR@TPR=95%), and TNR99 (TNR@TPR=99%). For detailed plots, we refer readers to Appendix B.3.

| Method | SVHN | | | LSUN | | |
|---|---|---|---|---|---|---|
| | AUROC (↑) | TNR95 (↑) | TNR99 (↑) | AUROC (↑) | TNR95 (↑) | TNR99 (↑) |
| SGLD | $0.784_{\pm0.031}$ | $0.536_{\pm0.039}$ | $0.424_{\pm0.034}$ | $0.853_{\pm0.015}$ | $0.520_{\pm0.038}$ | $0.276_{\pm0.039}$ |
| pSGLD | $0.745_{\pm0.009}$ | $0.506_{\pm0.012}$ | $0.408_{\pm0.018}$ | $0.855_{\pm0.009}$ | $0.520_{\pm0.025}$ | $0.255_{\pm0.043}$ |
| SGHMC | $\underline{0.790}_{\pm0.051}$ | $\underline{0.549}_{\pm0.068}$ | $0.427_{\pm0.030}$ | $0.860_{\pm0.018}$ | $0.554_{\pm0.028}$ | $0.357_{\pm0.034}$ |
| SGNHT | $0.776_{\pm0.014}$ | $0.530_{\pm0.013}$ | $\underline{0.436}_{\pm0.014}$ | $\underline{0.864}_{\pm0.009}$ | $\underline{0.556}_{\pm0.006}$ | $\underline{0.375}_{\pm0.015}$ |
| **PX-SGHMC (ours)** | $\mathbf{0.832}_{\pm0.014}$ | $\mathbf{0.632}_{\pm0.009}$ | $\mathbf{0.514}_{\pm0.021}$ | $\mathbf{0.884}_{\pm0.007}$ | $\mathbf{0.594}_{\pm0.037}$ | $\mathbf{0.405}_{\pm0.036}$ |

## 5.2 MAIN RESULTS: IMAGE CLASSIFICATION TASKS

### 5.2.1 CIFAR-10

We present results on CIFAR-10 (Krizhevsky et al., 2009) and extensively study in advanced tasks such as robustness analysis and OOD detection. For our experiments, we employ the R20-FRN-Swish architecture, representing ResNet20 with FRN normalization and Swish nonlinearity and adapted from the HMC checkpoints provided by Izmailov et al. (2021).

**Results on CIFAR.** Table 1 presents the evaluation on the CIFAR-10 test split in the 'C10' column. It shows that PX-SGHMC significantly outperforms other methods in terms of both ERR and NLL.

**Robustness to distribution shifts.** One of the key selling points of Bayesian methods is that they produce reliable predictions that account for uncertainty, leading us to evaluate robustness to distribution shifts (Recht et al., 2019; Taori et al., 2020; Miller et al., 2021). Specifically, we test on natural distribution shifts using CIFAR-like test datasets, including CIFAR-10.1 (Recht et al., 2019), CIFAR-10.2 (Lu et al., 2020), and STL (Coates et al., 2011), as well as on image corruptions using CIFAR-10-C (Hendrycks & Dietterich, 2019). Table 1 shows that our approach not only outperforms the baseline methods on the in-distribution data but also exhibits significant robustness to distribution shifts. Please refer to Appendix B.2 for further results on the CIFAR-10-C benchmark.

**Out-of-distribution detection.** Another important task for evaluating predictive uncertainty is the OOD detection. In real-world scenarios, models are likely to encounter random OOD examples, in which they are required to produce uncertain predictions (Hendrycks & Gimpel, 2017; Liang & Li, 2018). Categorical predictions of the classifiers are expected to be closer to being uniform for OOD inputs than for in-distribution ones. Therefore, we use predictive entropy (Lakshminarayanan et al., 2017) to distinguish between in-distribution examples from CIFAR-10 and OOD examples from SVHN (Netzer et al., 2011) and LSUN (Yu et al., 2015). Table 2 demonstrates our method

**Table 3: Results with data augmentation.** Evaluation results on C10 (CIFAR-10), C100 (CIFAR-100), and TIN (TinyImageNet) with data augmentation. In this context, we manually set the posterior temperature to 0.01 to account for the increased data resulting from augmentation.

| Method | ERR (↓) | | | NLL (↓) | | |
|---|---|---|---|---|---|---|
| | C10 | C100 | TIN | C10 | C100 | TIN |
| SGLD | $0.080_{\pm0.002}$ | $0.326_{\pm0.006}$ | $0.546_{\pm0.004}$ | $0.246_{\pm0.004}$ | $1.180_{\pm0.018}$ | $2.278_{\pm0.010}$ |
| pSGLD | $0.097_{\pm0.002}$ | $0.412_{\pm0.007}$ | $0.601_{\pm0.005}$ | $0.306_{\pm0.004}$ | $1.546_{\pm0.035}$ | $2.562_{\pm0.015}$ |
| SGHMC | $\underline{0.071}_{\pm0.001}$ | $\underline{0.319}_{\pm0.002}$ | $\underline{0.538}_{\pm0.003}$ | $\underline{0.223}_{\pm0.002}$ | $\underline{1.138}_{\pm0.008}$ | $2.251_{\pm0.022}$ |
| SGNHT | $0.074_{\pm0.001}$ | $0.335_{\pm0.004}$ | $\underline{0.536}_{\pm0.002}$ | $0.231_{\pm0.006}$ | $1.199_{\pm0.014}$ | $\underline{2.240}_{\pm0.013}$ |
| **PX-SGHMC (ours)** | $\mathbf{0.069}_{\pm0.001}$ | $\mathbf{0.290}_{\pm0.004}$ | $\mathbf{0.498}_{\pm0.004}$ | $\mathbf{0.217}_{\pm0.005}$ | $\mathbf{1.030}_{\pm0.011}$ | $\mathbf{2.089}_{\pm0.008}$ |

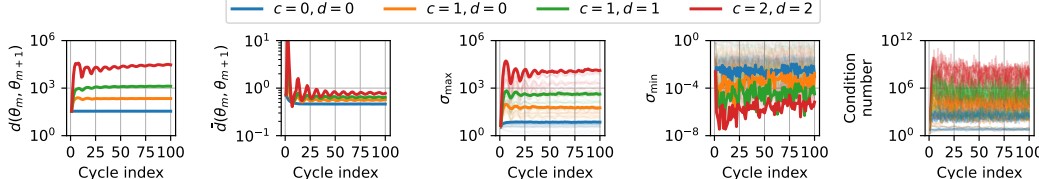

**Figure 2: Connection between exploration and singular value dynamics.** The first and second plots illustrate exploration through unnormalized and normalized Euclidean distances, while the third to fifth plots depict singular value dynamics, represented by the largest and smallest singular values and condition numbers. For the 21 layers, the singular value plots feature 21 transparent lines for each item, with the maximum (or minimum) value highlighted as the representative.

shows greater predictive uncertainty in handling OOD examples, as indicated by metrics associated with the Receiver Operating Characteristic (ROC) curve.

### 5.2.2 RESULTS WITH DATA AUGMENTATIONS

Although our evaluations in Section 5.2.1 were conducted *without* data augmentation, practical settings typically involve it. Therefore, we present additional comparative results on various image classification datasets, including CIFAR-10, CIFAR-100, and TinyImageNet, using data augmentation that includes random cropping and horizontal flipping. Since these augmentations can lead to inaccurate likelihood estimation (Wenzel et al., 2020; Noci et al., 2021; Nabarro et al., 2022), we introduce the concept of a *cold posterior* in these experiments, i.e., setting $T < 1$ in Eq. 5 and sampling from the tempered posterior $p(\boldsymbol{\theta}|\mathcal{D})^{1/T}$.

Table 3 presents the results obtained by setting $T = 0.01$ in Eq. 7. It demonstrates that PX-SGHMC still outperforms the other methods in both ERR and NLL, indicating that the enhanced diversity is also applicable to practical scenarios involving data augmentations and posterior tempering across various datasets. Additionally, Appendix B.4 provides ablation results related to the cold posterior effect in the absence of data augmentation, showing that our method consistently outperforms the SGHMC baselines across varying temperatures.

### 5.2.3 EMPIRICAL ANALYSIS

**Connection between exploration and singular value dynamics.** Theorem 3.2 suggests that the update size of SGLD dynamics at each time step is constrained by the maximum singular value, implying that small singular values may limit exploration. In this regard, a key characteristic of EP in deep linear neural networks is the learning dynamics of singular values; Arora et al. (2019b) showed that the evolution rates of singular values are proportional to their magnitudes raised to the *power of* $2 - 2/e$, where $e$ represents the depth of the factorization. In other words, as the depth of matrices increases, larger singular values tend to grow, while smaller singular values shrinks close to zero. Although our setting does not fully align with the theoretical assumptions in Arora et al. (2019b), as we do not consider DLNNs, we empirically demonstrate the connection between exploration and singular value dynamics by varying the number of expanded matrices $e = c + d + 1$.

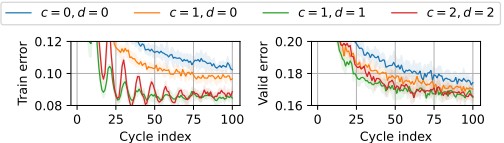

| Expansion | ERR ($\downarrow$) | NLL ($\downarrow$) | AMB ($\uparrow$) |
|---|---|---|---|
| $c = 0, d = 0$ | $0.135_{\pm0.005}$ | $0.444_{\pm0.007}$ | $0.196_{\pm0.004}$ |
| $c = 1, d = 0$ | $0.127_{\pm0.002}$ | $0.404_{\pm0.004}$ | $0.214_{\pm0.004}$ |
| $c = 1, d = 1$ | $\underline{0.116}_{\pm0.003}$ | $\mathbf{0.369}_{\pm0.005}$ | $\underline{0.253}_{\pm0.001}$ |
| $c = 2, d = 2$ | $\mathbf{0.114}_{\pm0.001}$ | $\underline{0.373}_{\pm0.003}$ | $\mathbf{0.278}_{\pm0.006}$ |

**Figure 3: Trace plots for EP.** It depicts training and validation errors along with trajectory.

**Table 4: Ablation results on EP.** Metrics are computed using the validation split.

Fig. 2 depicts our experimental findings: (i) The first and second plots quantify exploration by computing the Euclidean distance between consecutive posterior samples, defined as $d(\boldsymbol{\theta}_m, \boldsymbol{\theta}_{m+1}) = \|\boldsymbol{\theta}_{m+1} - \boldsymbol{\theta}_m\|_2$, where $\boldsymbol{\theta}_m$ denotes the sample at cycle index $m$. To exclude the effect of scale invariance in neural network parameters due to normalization layers such as FRN, we also compute their normalized version: $\bar{d}(\boldsymbol{\theta}_m, \boldsymbol{\theta}_{m+1}) = \|\boldsymbol{\theta}_{m+1} - \boldsymbol{\theta}_m\|_2 / \|\boldsymbol{\theta}_m\|_2$. Both unnormalized and normalized distances between consecutive samples increase as the number of expanded matrices $e = c + d + 1$ grows. (ii) The third and fourth plots illustrate the dynamics of singular values by plotting the maximum and minimum singular values of the convolution layer kernels. For each convolution layer, we compute the singular values of the kernel tensor following Sedghi et al. (2019). In line with the theoretical argument presented in Arora et al. (2019b), although our setup does not involve DLNNs, we observe that as the number of expanded matrices increases, the largest singular value rises while the smallest singular value decreases.

**EP converges faster than SP.** Another important property of EP in DLNNs is its accelerated convergence toward optima or modes (Arora et al., 2019a), which has also been observed in deep convolutional networks with nonlinearities (Guo et al., 2020). Building on this, we empirically investigate the local mode convergence of EP within the SGMCMC framework, where faster convergence is particularly crucial for BMA performance due to the slower local mode convergence caused by the injected Gaussian noise in SGMCMC methods (Zhang et al., 2020) compared to SGD in DE. Fig. 3 presents trace plots of training and validation errors, showing that both tend to converge more rapidly as the number of expanded matrices increases.

Based on the empirical findings, we hypothesize that EP induces a large maximum singular value, as shown in ii), which enlarges the upper bound in Theorem 3.2 and breaks the exploration limit, as demonstrated in i). Table 4 additionally presents the validation metrics for each setup and clearly demonstrates that the proposed EP indeed achieves better functional diversity, as indicated by the increased AMB. To sum up, PX-SGMCMC effectively enhances both the exploration and exploitation of a single SGHMC chain, resulting in improved BMA measured by ERR and NLL.

## 5.3 Comparative Analysis Using HMC Checkpoints

While Section 5.2 presents extensive experimental results by running SGMCMC algorithms from random initializations, we also conduct a comparison using HMC checkpoints provided by Izmailov et al. (2021) as an initialization of parameters in SGMCMC. Specifically, we run both SGHMC and PX-SGHMC starting from the burn-in checkpoint of HMC, employing hyperparameters aligned with those in Izmailov et al. (2021). Further experimental setups can be found in Appendix D.

### 5.3.1 Diversity Analysis

The diversity of the parameters does not necessarily imply that the diversity of the corresponding functions they represent; for instance, certain weight permutations and sign flips can leave the function invariant (Hecht-Nielsen, 1990; Chen et al., 1993). To effectively approximate the BMA integral in Eq. 3, functional diversity is essential (Wilson & Izmailov, 2020). Therefore, we quantify the diversity of posterior samples using their predictions by computing the ensemble ambiguity (Wood et al., 2023) as well as the variance of predictions (Ortega et al., 2022).

Table 5 clearly shows that PX-SGHMC exhibits (a) superior exploration in the parameter space compared to vanilla SGHMC, as evidenced by the average distances between consecutive samples, and (b) higher diversity in predictions, indicated by ensemble ambiguity and variance of predictions comparable to the gold standard HMC. Consequently, similar to HMC, (c) the BMA performance

**Table 5: Diversity analysis using HMC checkpoints.** We measure (a) parameter diversity using unnormalized and normalized Euclidean distances ($d$ and $\bar{d}$), (b) prediction diversity using ensemble ambiguity (AMB) and variance (VAR), and (c) individual (IND) and ensemble (ENS) negative log-likelihoods for 10 posterior samples from each method.

| Method | (a) | | (b) | | (c) | |
|---|---|---|---|---|---|---|
| | $d(\boldsymbol{\theta}_m, \boldsymbol{\theta}_{m+1})$ | $\bar{d}(\boldsymbol{\theta}_m, \boldsymbol{\theta}_{m+1})$ | AMB | VAR | IND | ENS |
| HMC | $322.8_{\pm 0.333}$ | $\mathbf{1.374}_{\pm 0.002}$ | **0.347** | **0.107** | 0.800 | 0.381 |
| SGHMC | $60.65_{\pm 0.258}$ | $0.258_{\pm 0.001}$ | 0.162 | 0.063 | **0.690** | 0.464 |
| **PX-SGHMC (Ours)** | $\mathbf{1290.}_{\pm 1.372}$ | $1.372_{\pm 0.150}$ | 0.339 | 0.105 | 0.739 | **0.353** |

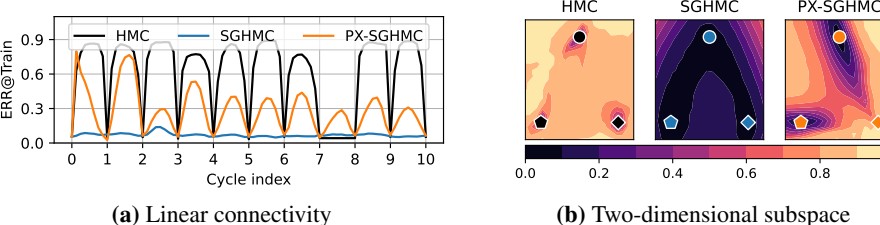

**(a)** Linear connectivity        **(b)** Two-dimensional subspace

**Figure 4: Loss landscape analysis using HMC checkpoints.** We visualize (a) linear connectivity between consecutive posterior samples and (b) a two-dimensional subspace spanned by the 0th (diamond), 1st (circle), and 2nd (pentagon) posterior samples. Both plots depict classification error on 1,000 training examples. Note that the 8th HMC sample was rejected and reverted to the 7th.

of PX-SGHMC surpasses that of SGHMC, although individual posterior samples exhibit worse performance in terms of negative log-likelihoods.

### 5.3.2 LOSS LANDSCAPE ANALYSIS

In this section, we investigate how effectively PX-SGHMC explores the posterior distribution over parameters compared to both HMC and SGHMC, from the perspective of loss surface geometry (Li et al., 2018). Fig. 4a visualizes the loss barrier between consecutive posterior samples, illustrating how often each method *jumps* over these barriers. While PX-SGHMC does not jump as high as HMC, the larger barriers it crosses compared to SGHMC indicate significantly better exploration, consistent with the discussion in Section 5.3. Fig. 4b visualizes a two-dimensional subspace spanning the right-most initial position (0th) and two subsequent posterior samples (1st and 2nd), demonstrating that the diversity of PX-SGHMC approaches that of the gold standard HMC when sampling from the multi-modal BNN posterior.

## 6 CONCLUSION

We have presented PX-SGMCMC, a simple yet effective parameter expansion technique tailored for SGMCMC, which decomposes each weight matrix in deep neural networks into the product of new expanded-parameter matrices. Our theoretical analysis shows that the proposed parameter expansion strategy provides a form of preconditioning on the gradient updates, enhancing the exploration of the posterior energy surface. The extensive experimental results strongly support our claims regarding the improved exploration linked to the singular value dynamics of the weight matrices explained in our theoretical analysis. As a result, the posterior samples obtained through PX-SGMCMC demonstrate increased diversity in both parameter and function spaces, comparable to the gold standard HMC, leading to improved predictive uncertainty and enhanced robustness to OOD data.

**Limitations.** While EP does not increase inference costs, it does require additional training resources in terms of memory and computation. In future work, we aim to optimize the reparameterization design to minimize these computational overheads while further enhancing diversity.

ETHICS STATEMENT

This paper does not raise any ethical concerns, as it presents a parameter expansion strategy based on the SGMCMC algorithm, which is free from ethical issues.

REPRODUCIBILITY STATEMENT

For the theoretical results in Section 3, we direct readers to Appendix A. All experimental details necessary for reproducibility can be found in Appendix D.

ACKNOWLEDGMENTS

This work was partly supported by Institute of Information & communications Technology Planning & Evaluation(IITP) grant funded by the Korea government(MSIT) (No.RS-2019-II190075, Artificial Intelligence Graduate School Program(KAIST); No.RS-2024-00509279, Global AI Frontier Lab), and the National Research Foundation of Korea(NRF) grant funded by the Korea government(MSIT) (NRF-2022R1A5A708390812; No. RS-2023-00279680). We also thank Byoungwoo Park and Hyungi Lee for their thoughtful discussion.

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

# A PROOFS

## A.1 PROOF OF LEMMA 3.1

*Proof.* The derivative of $\mathcal{F}$ with respect to $\mathbf{W}_j$ for every $j = 1, \ldots, e$ can be decomposed as

$$\frac{\partial \mathcal{F}}{\partial \mathbf{W}_j}(\mathbf{W}_1, \ldots, \mathbf{W}_e) = \mathbf{W}_{1:j-1}^\top \frac{\partial \mathcal{F}(\mathbf{W}_{1:e})}{\partial \mathbf{W}_{1:e}} \mathbf{W}_{j+1:e}^\top. \tag{18}$$

Substituting this in Eq. 12, we get

$$\frac{\mathrm{d}\mathbf{W}_j(t)}{\mathrm{d}t} = -\mathbf{W}_{1:j-1}^\top \frac{\partial \mathcal{F}(\mathbf{W}_{1:e}(t))}{\partial \mathbf{W}_{1:e}} \mathbf{W}_{j+1:e}^\top. \tag{19}$$

Therefore, assuming that $\mathbf{W}_{1:0} = \mathbf{W}_{e+1:1} = I$,

$$\frac{\mathrm{d}\mathbf{W}_{1:e}(t)}{\mathrm{d}t} = \sum_{j=1}^{e} \mathbf{W}_{1:j-1}(t) \frac{\mathrm{d}\mathbf{W}_j(t)}{\mathrm{d}t} \mathbf{W}_{j+1:e}(t) \tag{20}$$

$$= -\sum_{j=1}^{e} \mathbf{W}_{1:j-1}(t) \mathbf{W}_{1:j-1}(t)^\top \frac{\partial \mathcal{F}(\mathbf{W}_{1:e}(t))}{\partial \mathbf{W}_{1:e}} \mathbf{W}_{j+1:e}(t)^\top \mathbf{W}_{j+1:e}(t). \tag{21}$$

By taking the vectorization on both sides,

$$\mathrm{vec}\left(\frac{\mathrm{d}\mathbf{W}_{1:e}(t)}{\mathrm{d}t}\right)$$

$$= -\sum_{j=1}^{e} \left(\mathbf{W}_{j+1:e}(t)^\top \mathbf{W}_{j+1:e}(t) \otimes \mathbf{W}_{1:j-1}(t) \mathbf{W}_{1:j-1}(t)^\top\right) \mathrm{vec}\left(\frac{\partial \mathcal{F}(\mathbf{W}_{1:e}(t))}{\partial \mathbf{W}_{1:e}}\right) \tag{22}$$

$$= -P_{\mathbf{X}(t)} \nabla \mathcal{F}(\mathbf{X}(t)). \tag{23}$$

$\square$

## A.2 PROOF OF THEOREM 3.2

*Proof.* For

$$\mathcal{W}_j = \mathbf{W}_{j+1:e}^\top \mathbf{W}_{j+1:e} \otimes \mathbf{W}_{1:j-1} \mathbf{W}_{1:j-1}^\top \tag{24}$$

in Lemma 3.1, the precondition can be described as

$$P_{\mathbf{X}(t)} = \sum_{j=1}^{e} \mathcal{W}_j, \quad \mathbf{W}_{1:0} = \mathbf{W}_{e+1:e} = I. \tag{25}$$

Since $\mathcal{W}_j$ is positive semi-definite and symmetric, the singular values of $\mathcal{W}_j$ is the same as the absolute eigenvalues of $\mathcal{W}_j$. When $\mathbf{W}_{i:j} = U_{i:j} D_{i:j} V_{i:j}^\top$ by the singular value decomposition,

$$\mathbf{W}_{j+1:e}^\top \mathbf{W}_{j+1:e} \otimes \mathbf{W}_{1:j-1} \mathbf{W}_{1:j-1}^\top \tag{26}$$

$$= \left(V_{j+1:e} D_{j+1:e}^\top D_{j+1:e} V_{j+1:e}^\top\right) \otimes \left(U_{1:j-1} D_{1:j-1}^\top D_{1:j-1} U_{1:j-1}^\top\right) \tag{27}$$

$$= \left(V_{j+1:e} \otimes U_{1:j-1}\right) \left(D_{j+1:e}^\top D_{j+1:e} \otimes D_{1:j-1}^\top D_{1:j-1}\right) \left(V_{j+1:e} \otimes U_{1:j-1}\right)^\top \tag{28}$$

$$= O \Lambda O^\top \tag{29}$$

Therefore, the eigenvalues of $\mathcal{W}_j$ are

$$\sigma_r(\mathbf{W}_{j+1:e})^2 \sigma_{r'}(\mathbf{W}_{1:j-1})^2, \text{ for } r = 1, \ldots, n, \text{ and } r' = 1, \ldots, n, \tag{30}$$

where $\sigma_r$ is the $r$-th singular value. The min-max theorem for singular values yields

$$\sigma(\mathcal{W}_j) = \left|\sigma_r(\mathbf{W}_{j+1:e})^2 \sigma_{r'}(\mathbf{W}_{1:j-1})^2\right| \leq \prod_{i \neq j}^{e} \sigma_{\max}(\mathbf{W}_i)^2. \tag{31}$$

Using this value, we derive the upper bound from the fact that the operator $l_2$-norm of a matrix is the same as the maximum singular value. For $\mathbf{X} = \left(\text{vec}\left(\mathbf{W}_{1:e}^{(l)}\right)\right)_{l=1}^{L}$ such that $\mathbf{W}_{1:e} = \mathbf{W}_1\mathbf{W}_2\cdots\mathbf{W}_e$ and $\nabla\tilde{U}(\mathbf{X}(t)) = \nabla U(\mathbf{X}(t)) + \boldsymbol{s}$ such that $\mathbb{E}[\|\boldsymbol{s}\|] = s$, the distance between the two adjacent time steps is bounded as

$$\left\|\mathbf{X}^{(l)}(t+1) - \mathbf{X}^{(l)}(t)\right\|_2$$

$$= \epsilon\left\|-P_{\mathbf{X}^{(l)}(t)}\nabla\tilde{U}(\mathbf{X}^{(l)}(t)) + B_t\boldsymbol{\xi}\right\|_2 \tag{32}$$

$$= \epsilon\left\|-\sum_{j=1}^{e}\mathcal{W}_j\nabla\tilde{U}(\mathbf{X}^{(l)}(t)) + B_t\boldsymbol{\xi}^{(l)}\right\|_2 \tag{33}$$

$$\leq \epsilon\sum_{j=1}^{e}\|\mathcal{W}_j\|_2\left\|\nabla\tilde{U}(\mathbf{X}^{(l)}(t))\right\|_2 + \epsilon\left\|B_t\boldsymbol{\xi}^{(l)}\right\|_2 \tag{34}$$

$$\leq \epsilon\sum_{j=1}^{e}\prod_{i\neq j}^{e}\sigma_{\max}(\mathbf{W}_i^{(l)})\left(\left\|\nabla U(\mathbf{X}^{(l)}(t))\right\|_2 + \left\|\boldsymbol{s}^{(l)}\right\|_2\right) + \epsilon\left\|B_t\boldsymbol{\xi}^{(l)}\right\|_2. \tag{35}$$

Note that $\boldsymbol{\xi}$ is still the zero-mean Gaussian because we set the noise corresponding $\mathbf{W}_1,\ldots,\mathbf{W}_{j-1},\mathbf{W}_{j+1},\ldots,\mathbf{W}_e$ except for $\mathbf{W}_j$ zero. Once we take the all of layers and expectation on both sides,

$$\mathbb{E}\left[\|\mathbf{X}(t+1) - \mathbf{X}(t)\|\right]$$

$$\leq \epsilon\sum_{l=1}^{L}\mathbb{E}\left[\sum_{j=1}^{e}\prod_{i\neq j}^{e}\sigma_{\max}(\mathbf{W}_i^{(l)})\left(\|\nabla U(\mathbf{X}(t))\|_2 + \|\boldsymbol{s}\|_2\right) + \|B_t\boldsymbol{\xi}\|_2\right] \tag{36}$$

$$\leq \epsilon Le \cdot m^{(e-1)}(Lh + Ls) + \epsilon LC. \tag{37}$$

$\square$

# B SUPPLEMENTARY RESULTS

## B.1 EVALUATION METRICS

Let $\boldsymbol{z}_{m,i} \in \mathbb{R}^K$ represent the categorical logits predicted by the $m^{\text{th}}$ posterior sample $\boldsymbol{\theta}_s$ for the $i^{\text{th}}$ data point. The final ensemble prediction, which approximates the BMA integral of the predictive distribution, for the $i^{\text{th}}$ data point is given by:

$$\boldsymbol{p}_i = \frac{1}{M}\sum_{m=1}^{M}\boldsymbol{\sigma}(\boldsymbol{z}_{m,i}), \tag{38}$$

where $\boldsymbol{\sigma}$ denotes the softmax function, mapping categorical logits to probabilities. Using $\boldsymbol{p}_i$ and $\boldsymbol{z}_{m,i}$, as well as the ground truth label $y_i$ for the $i^{\text{th}}$ data point, we calculate the following evaluation metrics for a given set of $N$ data points.

**Classification error (ERR).** The classification error, often referred to as 0-1 loss, is the primary metric used to evaluate the performance of a classification model:

$$\text{ERR} = \frac{1}{N}\sum_{i=1}^{N}\left[y_i \neq \arg\max_k \boldsymbol{p}_i^{(k)}\right], \tag{39}$$

where $[\cdot]$ denotes the Iverson bracket.

**Negative log-likelihood (NLL).** The negative log-likelihood of a categorical distribution, commonly known as cross-entropy loss, serves as the key metric for assessing classification model performance in Bayesian literature:

$$\text{NLL} = \frac{1}{N}\sum_{i=1}^{N}\log \boldsymbol{p}_i^{(y_i)}. \tag{40}$$

**Table 6: Supplementary results for CIFAR-10-C.** It summarizes the classification error (ERR), negative log-likelihood (NLL), and expected calibration error (ECE) averaged over 19 corruption types for each intensity level. For a comprehensive overview of the results, we direct readers to Fig. 5, which illustrates the box-and-whisker plots.

| Metric | Method | AVG | Intensity level 1 | 2 | 3 | 4 | 5 |
|---|---|---|---|---|---|---|---|
| ERR ($\downarrow$) | SGLD | $0.301_{\pm0.137}$ | $0.206_{\pm0.079}$ | $0.253_{\pm0.091}$ | $0.294_{\pm0.115}$ | $0.341_{\pm0.144}$ | $\underline{0.410}_{\pm0.153}$ |
| | pSGLD | $0.317_{\pm0.131}$ | $0.216_{\pm0.076}$ | $0.266_{\pm0.081}$ | $0.309_{\pm0.101}$ | $0.360_{\pm0.130}$ | $0.433_{\pm0.142}$ |
| | SGHMC | $\underline{0.294}_{\pm0.140}$ | $\underline{0.194}_{\pm0.082}$ | $0.242_{\pm0.091}$ | $\underline{0.284}_{\pm0.113}$ | $\underline{0.335}_{\pm0.143}$ | $0.414_{\pm0.157}$ |
| | SGNHT | $0.296_{\pm0.136}$ | $0.195_{\pm0.077}$ | $\underline{0.241}_{\pm0.084}$ | $0.286_{\pm0.106}$ | $0.339_{\pm0.134}$ | $0.421_{\pm0.150}$ |
| | **PX-SGHMC (ours)** | $\mathbf{0.275}_{\pm0.126}$ | $\mathbf{0.180}_{\pm0.073}$ | $\mathbf{0.224}_{\pm0.081}$ | $\mathbf{0.264}_{\pm0.098}$ | $\mathbf{0.315}_{\pm0.120}$ | $\mathbf{0.394}_{\pm0.134}$ |
| NLL ($\downarrow$) | SGLD | $0.894_{\pm0.397}$ | $0.623_{\pm0.210}$ | $0.748_{\pm0.235}$ | $0.863_{\pm0.310}$ | $1.006_{\pm0.410}$ | $1.229_{\pm0.475}$ |
| | pSGLD | $0.945_{\pm0.383}$ | $0.659_{\pm0.203}$ | $0.792_{\pm0.217}$ | $0.912_{\pm0.278}$ | $1.066_{\pm0.374}$ | $1.298_{\pm0.447}$ |
| | SGHMC | $\underline{0.877}_{\pm0.420}$ | $\underline{0.585}_{\pm0.222}$ | $\underline{0.715}_{\pm0.244}$ | $\underline{0.837}_{\pm0.317}$ | $\underline{0.994}_{\pm0.419}$ | $1.253_{\pm0.502}$ |
| | SGNHT | $0.878_{\pm0.393}$ | $\underline{0.585}_{\pm0.203}$ | $\underline{0.715}_{\pm0.223}$ | $0.842_{\pm0.294}$ | $0.995_{\pm0.381}$ | $\underline{1.252}_{\pm0.450}$ |
| | **PX-SGHMC (ours)** | $\mathbf{0.826}_{\pm0.371}$ | $\mathbf{0.550}_{\pm0.194}$ | $\mathbf{0.673}_{\pm0.220}$ | $\mathbf{0.784}_{\pm0.276}$ | $\mathbf{0.934}_{\pm0.347}$ | $\mathbf{1.189}_{\pm0.420}$ |
| ECE ($\downarrow$) | SGLD | $0.082_{\pm0.061}$ | $0.070_{\pm0.023}$ | $0.066_{\pm0.024}$ | $0.074_{\pm0.045}$ | $0.094_{\pm0.071}$ | $0.107_{\pm0.099}$ |
| | pSGLD | $0.074_{\pm0.047}$ | $0.074_{\pm0.019}$ | $0.060_{\pm0.023}$ | $\mathbf{0.059}_{\pm0.035}$ | $0.077_{\pm0.056}$ | $\underline{0.100}_{\pm0.071}$ |
| | SGHMC | $0.076_{\pm0.062}$ | $\mathbf{0.064}_{\pm0.023}$ | $\underline{0.058}_{\pm0.024}$ | $0.064_{\pm0.045}$ | $0.081_{\pm0.072}$ | $0.111_{\pm0.098}$ |
| | SGNHT | $\underline{0.072}_{\pm0.053}$ | $\mathbf{0.064}_{\pm0.017}$ | $\mathbf{0.057}_{\pm0.020}$ | $0.060_{\pm0.038}$ | $\underline{0.073}_{\pm0.062}$ | $0.105_{\pm0.084}$ |
| | **PX-SGHMC (ours)** | $\mathbf{0.066}_{\pm0.031}$ | $\underline{0.067}_{\pm0.017}$ | $0.059_{\pm0.021}$ | $\mathbf{0.059}_{\pm0.019}$ | $\mathbf{0.063}_{\pm0.031}$ | $\mathbf{0.084}_{\pm0.049}$ |

**Ensemble ambiguity (AMB).** The generalized ambiguity decomposition for the cross-entropy loss is given by (Wood et al., 2023):

$$\text{AMB} = \underbrace{\frac{1}{M}\sum_{m=1}^{M}\frac{1}{N}\sum_{i=1}^{N}\log\boldsymbol{\sigma}(\boldsymbol{z}_{m,i})^{(y)}}_{\text{average loss}} - \underbrace{\frac{1}{N}\sum_{i=1}^{N}\log\boldsymbol{\sigma}\left(\frac{1}{M}\sum_{m=1}^{M}\boldsymbol{z}_{m,i}\right)^{(y)}}_{\text{ensemble loss}}. \tag{41}$$

Notably, logit ensembling in the ensemble loss term is essentially equivalent to computing a normalized geometric mean of the categorical probabilities. See Wood et al. (2023) for more details.

**Expected calibration error (ECE).** The expected calibration error with binning is a widely used metric for assessing the calibration of categorical predictions (Naeini et al., 2015):

$$\text{ECE} = \sum_{j=1}^{J}\frac{|B_j|\cdot|\text{acc}(B_j) - \text{conf}(B_j)|}{N}, \tag{42}$$

where $B_j$ represents the $j^{\text{th}}$ bin that includes $|B_j|$ data points with prediction confidence $\max_k \boldsymbol{p}_i^{(k)}$ falling within the interval $((j-1)/J, j/J]$. Here, $\text{acc}(B_j)$ indicates the classification accuracy, while $\text{conf}(B_j)$ refers to the average prediction confidence within the $j^{\text{th}}$ bin.

### B.2 ROBUSTNESS TO COMMON CORRUPTION

Table 6 presents the classification error and uncertainty metrics, including negative log-likelihood and expected calibration error (Naeini et al., 2015), for each level of corruption intensity. Our PX-SGHMC consistently outperforms all baseline methods across all metrics, with the number of bins for computing expected calibration error set to 15. Notably, PX-SGHMC shows lower calibration error with increasing intensity levels, demonstrating enhanced robustness to more severely corrupted inputs. Fig. 5 further presents box-and-whisker plots illustrating metrics across 19 corruption types for five intensity levels. Overall, PX-SGHMC exhibits better calibration than the other methods.

### B.3 OUT-OF-DISTRIBUTION DETECTION

To obtain the ROC curve and associated metrics (i.e., AUROC and TNR at TPR of 95% and 99%, as shown in Table 2), we used 1,000 in-distribution (ID) examples as positives and 1,000 out-of-distribution (OOD) examples as negatives. We manually balanced the number of examples, because

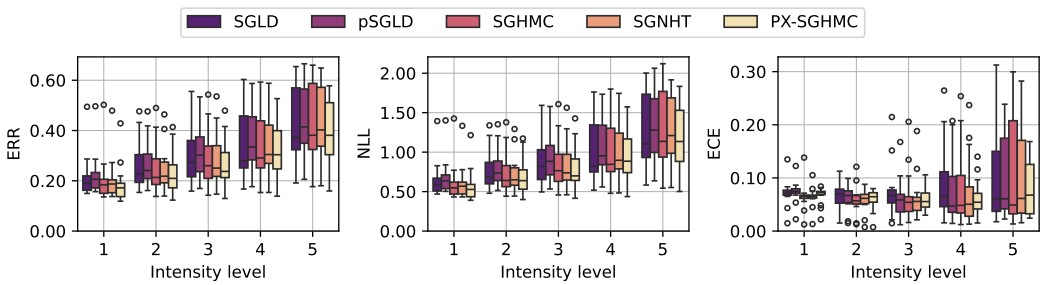

**Figure 5: Supplementary box-and-whisker plots for CIFAR-10-C.** It illustrates the classification error (ERR), negative log-likelihood (NLL), and expected calibration error (ECE) across 19 corruption types for five intensity levels.

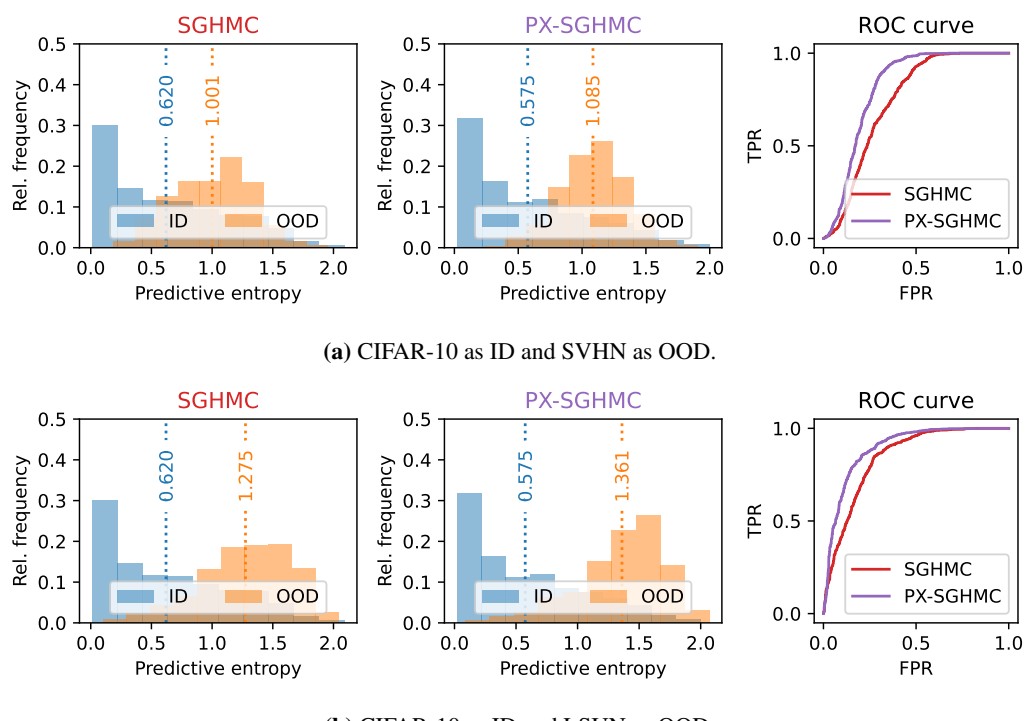

**(a)** CIFAR-10 as ID and SVHN as OOD.

**(b)** CIFAR-10 as ID and LSUN as OOD.

**Figure 6: Supplementary plots for out-of-distribution detection.** Histograms of predictive entropy for ID and OOD datasets, along with the receiver operating characteristic (ROC) curve measuring the separability between ID and OOD.

the ROC curve becomes less reliable when there is an imbalance between positive and negative examples. To see the perceptual differences between ID (CIFAR-10) and OOD (SVHN and LSUN) images, please refer to Fig. 10.

In Fig. 6, histograms show how our PX-SGHMC more effectively assigns low predictive entropy to ID inputs and high entropy to OOD inputs compared to the baseline (with SGHMC as a representative), while ROC curves assess the separability between the ID and OOD histograms. It clearly shows that PX-SGHMC is more robust to OOD inputs, offering more reliable predictions when encountering OOD inputs in real-world scenarios.

### B.4 ADDITIONAL RESULTS WITH COLD POSTERIOR

To obtain valid posterior samples from $p(\boldsymbol{\theta}|\mathcal{D})$, the temperature should be one in Langevin dynamics and its practical discretized implementations (i.e., $T = 1$ in Eqs. 5 and 7). However, many works in the Bayesian deep learning literature have, in practice, considered using $T < 1$, which is called *cold posterior* (Wenzel et al., 2020). Therefore, we further present comparative results between SGHMC and PX-SGHMC using the cold posterior.

Fig. 7 presents the evaluation results on CIFAR-10, CIFAR-10.1, CIFAR-10.2, STL, and CIFAR-10-C, as in Table 1. It is clear that our PX-SGHMC consistently outperforms the SGHMC baseline across all datasets and temperature values considered. These results suggest that our EP strategy functions orthogonally to the modification of the target posterior through posterior tempering.

### B.5 ADDITIONAL RESULTS ON ACCEPTANCE PROBABILITY OF GGMC

While SGHMC omits the Metropolis-Hastings correction, Garriga-Alonso & Fortuin (2021) recently argued that SGHMC effectively has an acceptance probability of zero, as the backward trajectory is not realizable in practice due to discretization. Expanding on this, they revisited Gradient-Guided Monte Carlo (GGMC; Horowitz, 1991), a method that generalizes HMC and SGLD while ensuring a positive acceptance probability. Building on their insights, we further investigate how the proposed EP influences the acceptance probability within the GGMC framework.

Fig. 8 illustrates the following for both EP and SP:

- As the peak step size $\epsilon_0$ increases, GGMC produces more diverse samples, as indicated by the unnormalized Euclidean distances in (a). This diversity ultimately contributes to improved ensemble predictions, as shown in (c).

- However, as the peak step size increases, the simulation error also grows. A step size of $\epsilon_0 = 3 \times 10^{-4}$, which yields good performance in practical applications, results in a relatively low acceptance probability of around 25%.

Notably, the proposed EP enhances both exploration and simulation. The implicit step size scaling introduced by EP facilitates improved exploration without compromising the acceptance probability due to discretization effects–indeed, it may even enhance it. This indicates that the superior exploration capability achieved by EP cannot be replicated solely by increasing the step size.

### B.6 COST ANALYSIS AND LOW-RANK VARIANT

We have compiled system logs comparing PX-SGHMC with SGHMC in Table 7. Notably, the logs show that PX-SGHMC exhibits no significant differences both in sampling speed and memory consumption compared to SGHMC in practice.

Moreover, we further implemented a low-rank variant of our approach which reduces memory overhead. Specifically, we used a low-rank plus diagonal approach to construct the expanded matrices, i.e., we compose a new expanded matrix $\mathbf{P} = \mathbf{D} + \mathbf{L}_1^\top \mathbf{L}_2$ for a diagonal matrix $\mathbf{D} \in \mathbb{R}^{p \times p}$ and the low-rank matrices $\mathbf{L}_1, \mathbf{L}_2 \in \mathbb{R}^{r \times p}$ with $r < p$, which reduces the memory from $O(p^2)$ to $O(p + 2pr)$. In Table 8, we set $r$ to $p/8$ and $p/4$ for the $c = d = 1$ setup, resulting in 303,610 and 330,874 parameters during the sampling procedure, respectively. Even with the expanded matrices of the low-rank plus diagonal form, PX-SGHMC continues to outperform SGHMC, highlighting a clear direction for effectively addressing the increased parameter count of our method. Therefore, the design of expanded parameters is left to users with limited memory resources.

### B.7 ABLATION RESULTS ON BURN-IN PERIOD

In our main experiments, none of the SGMCMC methods utilize a separate burn-in phase (unlike Izmailov et al. (2021)); in other words, even samples from the first cycle are included in the BMA computation. To further analyze convergence, we conducted an additional ablation study on burn-in, following Izmailov et al. (2021), by examining the performance of BMA estimates and individual samples as a function of burn-in length. In Fig. 9, the x-axis represents the number of burn-in samples, i.e., the length of the burn-in period, while the y-axis shows the individual performance

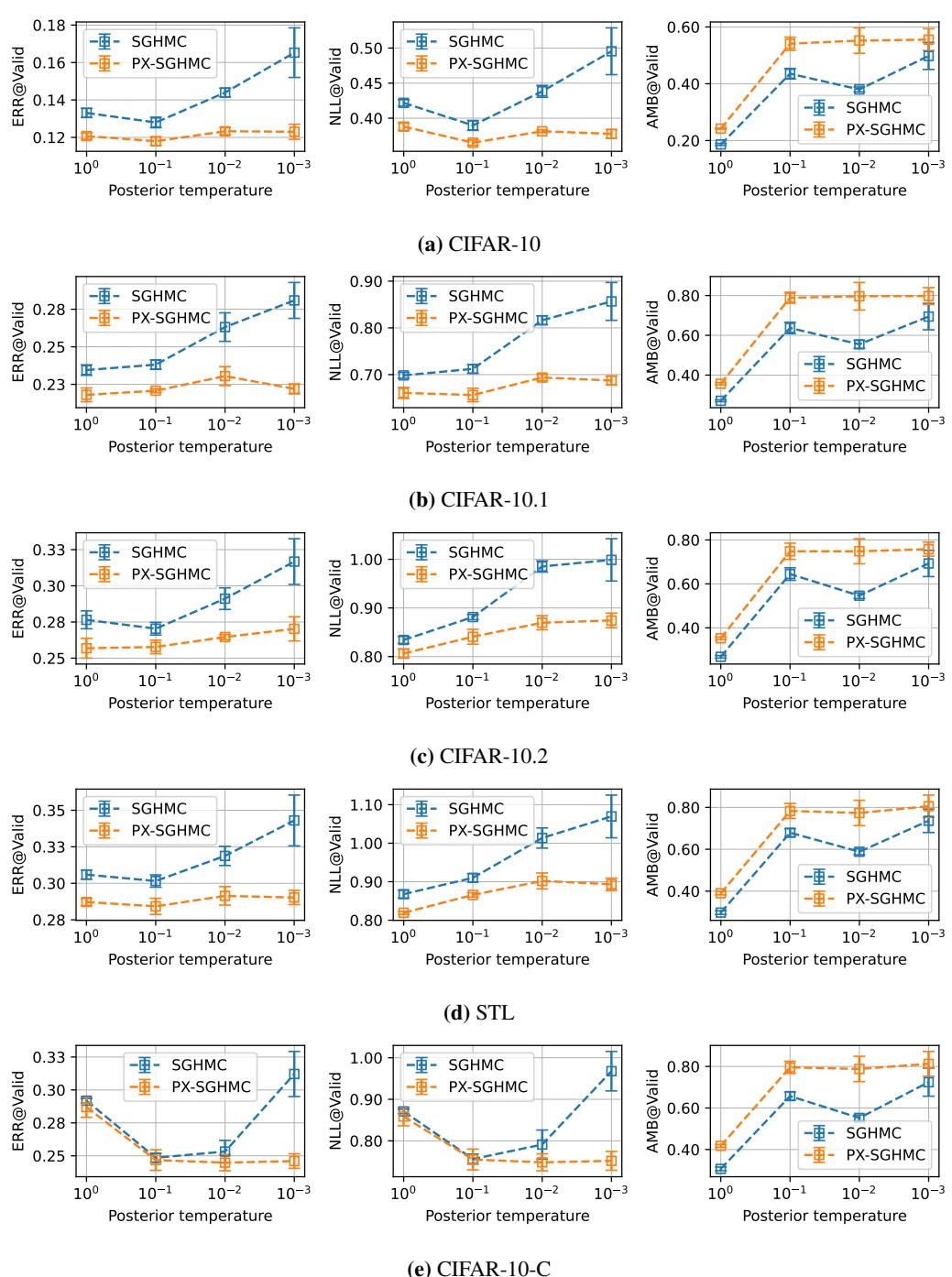

**Figure 7: Additional results with varying posterior temperature.** It depicts evaluation results for SGHMC and PX-SGHMC with cold posterior.

of the 50 samples (in average) obtained after the burn-in period (denoted as "IND (1)") and the ensemble performance (denoted as "BMA (50)"). For reference, we also plotted the performance using 100 samples without a burn-in period, previously reported in the main text (denoted as "BMA (100)"). It clearly demonstrates the the enhanced training dynamics introduced by our EP enable higher BMA performance with a shorter burn-in period. In other words, when collecting the same

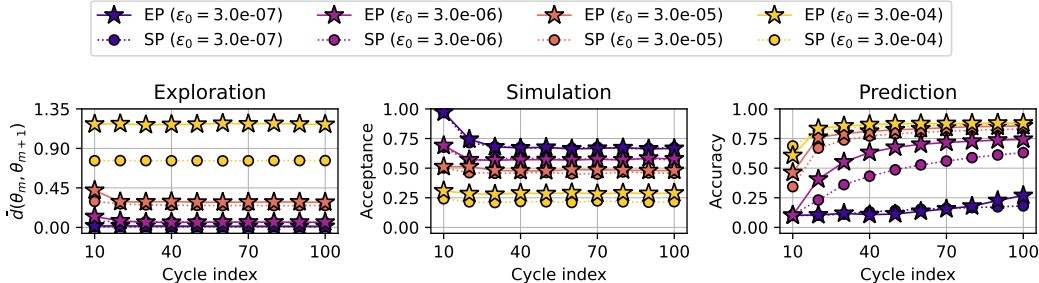

**Figure 8: Additional results using GGMC.** Trace plots illustrating **(a) Exploration (higher is better)**, which measures the normalized Euclidean distance from the previous sample; **(b) Simulation (higher is better)**, which represents the acceptance ratio of the sample; and **(c) Prediction (higher is better)**, which assesses the final performance of the ensemble prediction.

**Table 7: Cost analysis of SGHMC and PX-SGHMC.** It summarizes the costs involved in the sampling procedure for SGHMC and PX-SGHMC with $c = d = 1$ and $c = d = 2$. "Space (in theory)" refers to the number of parameters during the sampling process, while "Space (in practice)" represents the actual GPU memory allocated in our experimental setup using a single RTX A6000. "Time (in practice)" denotes the wall-clock time for each cycle, consisting of 5,000 steps.

| Method | Space (in theory) | Space (in practice) | Time (in practice) |
|---|---|---|---|
| SGHMC | 274,042 | 1818 MB | 61 sec/cycle |
| PX-SGHMC ($c = d = 1$) | 383,098 | 1838 MB | 64 sec/cycle |
| PX-SGHMC ($c = d = 2$) | 492,154 | 1852 MB | 73 sec/cycle |

50 samples, SGHMC requires a much longer burn-in period to achieve performance comparable to that of PX-SGHMC.

### B.8 COMPARATIVE RESULTS WITH META-LEARNING APPROACH

We further provide comparative results with the Learning to Explore (L2E; Kim et al., 2024) method. Since they also adopted the experimental setup of Izmailov et al. (2021) using the R20-FRN-Swish architecture, the results are largely comparable when key experimental configurations–such as data augmentation, the number of steps per cycle, and the number of posterior samples–are aligned.

Using the official implementation of Kim et al. (2024)[1], we ran the L2E method using the same setup as in Table 3 of main text, i.e., with data augmentation, cold posterior with $T = 0.01$, 5000 steps per cycle, and 100 posterior samples. Table 9 summarizes the results. Notably, our PX-SGHMC, which applies a vanilla SGHMC sampler to the expanded parameterization, yield competitive results compared to L2E, despite the latter relying on a more resource-intensive meta-learned sampler.

### B.9 ABLATION RESULTS ON STEP SIZE SCHEDULE

In our main experiments, the cyclical step size schedule is used for all SGMCMC methods. Since the cyclical step size schedule itself is designed to enhance exploration in SGMCMC and improve sample diversity (Zhang et al., 2020), we conducted ablation experiments on SP/EP parameterizations and constant/cyclical step size schedules to more clearly isolate the contribution of EP.

Table 10 summarizes the performance (ERR, NLL; lower is better) and functional diversity (AMB; higher is preferred) of SGHMC and PX-SGHMC under both constant and cyclical step size schedules. Based on "SGHMC w/ constant schedule," the results clearly show that our expanded parameterization (EP) contributes more significantly to performance improvements than the adoption of the cyclical schedule (CS).

---

[1]https://github.com/ciao-seohyeon/l2e

**Table 8: Low-rank variant of PX-SGHMC.** It summarizes the evaluation results for low-rank variants of PX-SGHMC with $c = d = 1$. "# Params (sampling)" indicates the number of parameters during the sampling process, while "# Params (inferece)" refers to the number of parameters after merging expanded matrices into the base matrix.

| | Memory costs | | Evaluation metrics | | |
|---|---|---|---|---|---|
| Method | # Params (sampling) | # Params (inference) | ERR ($\downarrow$) | NLL ($\downarrow$) | AMB ($\uparrow$) |
| SGHMC | 274,042 (x1.00) | 274,042 (x1.00) | 0.131 | 0.421 | 0.183 |
| PX-SGHMC ($r = p/8$) | 303,610 (x1.11) | 274,042 (x1.00) | 0.126 | 0.401 | 0.210 |
| PX-SGHMC ($r = p/4$) | 330,874 (x1.21) | 274,042 (x1.00) | **0.122** | **0.385** | 0.215 |
| PX-SGHMC ($r = p$) | 383,098 (x1.40) | 274,042 (x1.00) | 0.123 | 0.396 | **0.242** |

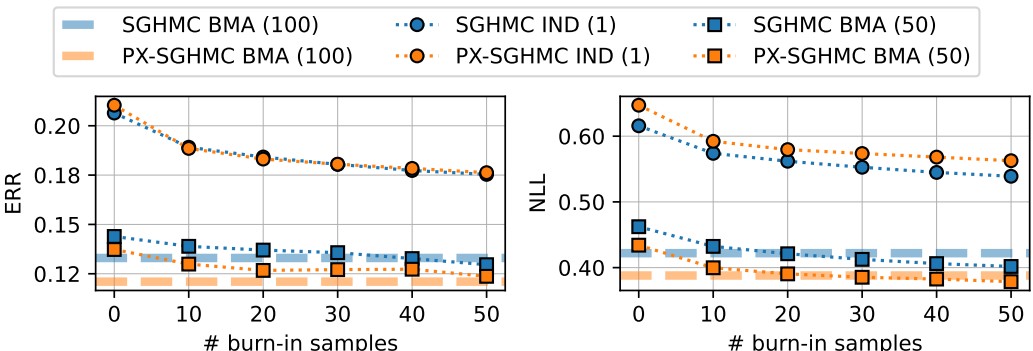

**Figure 9: Convergence of SGHMC and PX-SGHMC.** Performance comparison of individual posterior samples (IND) and Bayesian model averaging (BMA) ensembles with 50 samples from each SGHMC and PX-SGHMC chain, evaluated as a function of burn-in length. "BMA (100)" represents a burn-in length of 0 with a BMA ensemble of 100 samples, as used in the main experiments of this paper. "IND (1)" and "BMA (50)" refer to average individual sample performance and ensemble performance of 50 samples, respectively.

## C  ALGORITHMS

In this section, we outline the practical implementation of the SGMCMC algorithms used in our experiments: Stochastic Gradient Langevin Dynamics (SGLD; Welling & Teh, 2011), Stochastic Gradient Hamiltonian Monte Carlo (SGHMC; Chen et al., 2014), Stochastic Gradient Nosé-Hoover Thermostat (SGNHT; Ding et al., 2014), and preconditioned SGLD (pSGLD; Li et al., 2016). Additionally, we experimented with Stochastic Gradient Riemann Hamiltonian Monte Carlo (SGRHMC; Ma et al., 2015) using diagonal empirical Fisher and RMSProp estimates for the preconditioner. However, within the hyperparameter range explored, it demonstrated significantly lower performance than SGLD, leading us to exclude it from further experiments.

First, we present the hyperparameters that are consistent across all algorithms:

- $M$: the number of sampling cycles, representing the total number of posterior samples.

- $T$: the number of updates per cycle, resulting in $C \times T$ total updates.

- $\epsilon_t$: the step size at time step $t$. It can follow a 'Cyclical' with peak learning rate of $\epsilon_0$, defined as $\epsilon_t = \frac{\epsilon_0}{2} \left[ \cos \left( \frac{\pi \bmod(t,T)}{T} \right) + 1 \right]$, or remain 'Constant', i.e., $\forall t : \epsilon_t = \epsilon_0$.

- $\sigma^2$: the variance of the zero-mean Gaussian prior over the neural network parameters, where $p(\boldsymbol{\theta}) = \mathcal{N}(\boldsymbol{\theta}; \sigma^2 \mathbf{I})$.

- $\mathcal{B}_{m,t}$: the mini-batch at the $t^{\text{th}}$ time step of the $m^{\text{th}}$ cycle, with a size of $|\mathcal{B}|$.

**Table 9: Comparative results with data augmentation.** Evaluation results on C10 (CIFAR-10), C100 (CIFAR-100), and TIN (TinyImageNet) with data augmentation. In this context, we manually set the posterior temperature to $0.01$ to account for the increased data resulting from augmentation.

| | ERR ($\downarrow$) | | | NLL ($\downarrow$) | | |
|---|---|---|---|---|---|---|
| Method | C10 | C100 | TIN | C10 | C100 | TIN |
| SGHMC | $0.071_{\pm0.001}$ | $0.319_{\pm0.002}$ | $0.538_{\pm0.003}$ | $0.223_{\pm0.002}$ | $1.138_{\pm0.008}$ | $2.251_{\pm0.022}$ |
| PX-SGHMC (ours) | $\mathbf{0.069}_{\pm0.001}$ | $0.290_{\pm0.004}$ | $0.498_{\pm0.004}$ | $\mathbf{0.217}_{\pm0.005}$ | $1.030_{\pm0.011}$ | $2.089_{\pm0.008}$ |
| L2E (Kim et al., 2024) | $0.071_{\pm0.001}$ | $\mathbf{0.268}_{\pm0.002}$ | $\mathbf{0.488}_{\pm0.002}$ | $0.232_{\pm0.002}$ | $\mathbf{0.980}_{\pm0.006}$ | $\mathbf{2.062}_{\pm0.011}$ |

**Table 10: Additional results with a constant step size.** Classification error (ERR), negative log-likelihood (NLL), and ensemble ambiguity (AMB) on the test split of CIFAR-10, comparing the use of the cyclical schedule (CS) and our proposed expanded parameterization (EP).

| | Components | | Evaluation metrics | | |
|---|---|---|---|---|---|
| Label | CS | EP | ERR ($\downarrow$) | NLL ($\downarrow$) | AMB ($\uparrow$) |
| SGHMC w/ constant schedule | | | $0.135_{\pm0.003}$ | $0.441_{\pm0.003}$ | $0.209_{\pm0.001}$ |
| SGHMC w/ cyclical schedule | ✓ | | $0.133_{\pm0.002}$ | $0.422_{\pm0.005}$ | $0.186_{\pm0.004}$ |
| PX-SGHMC w/ constant schedule | | ✓ | $\mathbf{0.115}_{\pm0.002}$ | $\mathbf{0.379}_{\pm0.002}$ | $0.232_{\pm0.003}$ |
| PX-SGHMC w/ cyclical schedule | ✓ | ✓ | $0.121_{\pm0.002}$ | $0.388_{\pm0.005}$ | $\mathbf{0.242}_{\pm0.002}$ |

Next, we briefly summarize the additional components introduced in each method. For a more in-depth exploration of SGMCMC methods, we refer readers to Ma et al. (2015) and references therein, which offer a concise summary from the perspective of stochastic differential equations.

- pSGLD introduces adaptive preconditioners from optimization, e.g., RMSProp.

- SGHMC introduces the friction matrix $\mathbf{C}$ to mitigate the noise from mini-batch gradients. While the friction term is originally a matrix, it is often implemented as a scalar value in practice ($\mathbf{C} = C\mathbf{I}$). Also, the gradient noise estimate is set to zero ($\hat{\mathbf{B}} = \mathbf{0}$), and the mass matrix is defined as the identity matrix ($\mathbf{M} = \mathbf{I}$) in practical implementations.

- SGNHT introduces an auxiliary thermostat variable $\xi$ to maintain thermal equilibrium of the system. In practical implementations, it can be interpreted as making the friction term used for momentum decay in SGHMC learnable. Intuitively, when the mean kinetic energy exceeds $1/2$, $\xi$ increases, leading to greater friction on the momentum.

Algorithms 1, 3 and 4 summarize our practical implementations of SGLD, pSGLD, SGHMC, and SGNHT, while Appendix B provides a detailed hyperparameter setup for each method used in our experiments.

# D    Experimental Details

## D.1    Software and Hardware

We built our experimental code using JAX (Bradbury et al., 2018), which is licensed under Apache-2.0.[2] All experiments were conducted on machines equipped with an RTX 2080, RTX 3090, or RTX A6000. The code is available at https://github.com/cs-giung/px-sgmcmc.

## D.2    Image Classification on CIFAR

**Dataset.** CIFAR-10 (Krizhevsky et al., 2009) is a dataset comprising $32 \times 32 \times 3$ images classified into 10 categories. We utilized 40,960 training examples and 9,040 validation examples based on the HMC settings from Izmailov et al. (2021), with the final evaluation conducted on 10,000 test

---

[2]https://www.apache.org/licenses/LICENSE-2.0

---

**Algorithm 1:** Practical implementation of SGLD

**Require:** the hyperparameters mentioned above.
**Ensure:** a set of posterior samples $\Theta \leftarrow \{\}$.

Initialize position $\boldsymbol{\theta}_{0,T}$ from scratch or pre-set values.
**for** $m = 1, 2, \ldots, M$ **do**
    Initialize $\boldsymbol{\theta}_{m,0} \leftarrow \boldsymbol{\theta}_{m-1,T}$.
    **for** $t = 1, 2, \ldots, T$ **do**
        $\boldsymbol{z}_{m,t} \sim \mathcal{N}(\mathbf{0}, \mathbf{I})$.
        $\boldsymbol{g}_{m,t} = \nabla_{\boldsymbol{\theta}} \tilde{U}(\boldsymbol{\theta}; \mathcal{B}_{m,t})|_{\boldsymbol{\theta}=\boldsymbol{\theta}_{m,t-1}}$.
        $\boldsymbol{\theta}_{m,t} = \boldsymbol{\theta}_{m,t-1} - \epsilon_t \boldsymbol{g}_{m,t} + \sqrt{2\epsilon_t} \boldsymbol{z}_{m,t}$.
    **end**
    Collect $\Theta \leftarrow \Theta \cup \{\boldsymbol{\theta}_{m,T}\}$.
**end**
**return** $\Theta$

---

**Algorithm 2:** Practical implementation of pSGLD

**Require:** smoothing factor $\beta$ and the hyperparameters mentioned above.
**Ensure:** a set of posterior samples $\Theta \leftarrow \{\}$.

Initialize position $\boldsymbol{\theta}_{0,T}$ from scratch or pre-set values.
**for** $m = 1, 2, \ldots, M$ **do**
    Initialize $\boldsymbol{\theta}_{m,0} \leftarrow \boldsymbol{\theta}_{m-1,T}$.
    **for** $t = 1, 2, \ldots, T$ **do**
        $\boldsymbol{z}_{m,t} \sim \mathcal{N}(\mathbf{0}, \mathbf{I})$.
        $\boldsymbol{g}_{m,t} = \nabla_{\boldsymbol{\theta}} \tilde{U}(\boldsymbol{\theta}; \mathcal{B}_{m,t})|_{\boldsymbol{\theta}=\boldsymbol{\theta}_{m,t-1}}$.
        $\boldsymbol{\nu}_{m,t} = \beta \boldsymbol{\nu}_{m,t-1} + (1-\beta) \boldsymbol{g}_{m,t}^{\circ 2}$.
        $\boldsymbol{\theta}_{m,t} = \boldsymbol{\theta}_{m,t-1} - \epsilon_t \boldsymbol{g}_{m,t} \oslash (\boldsymbol{\nu}_{m,t}^{\circ 1/2} + \varepsilon) + \sqrt{2\epsilon_t} \boldsymbol{z}_{m,t} \oslash (\boldsymbol{\nu}_{m,t}^{\circ 1/2} + \varepsilon)^{\circ 1/2}$.
    **end**
    Collect $\Theta \leftarrow \Theta \cup \{\boldsymbol{\theta}_{m,T}\}$.
**end**
**return** $\Theta$

---

examples. For a comprehensive evaluation, we also employed additional datasets, including STL-10 (Coates et al., 2011), CIFAR-10.1 (Recht et al., 2019), CIFAR-10.2 (Lu et al., 2020), and CIFAR-10-C (Hendrycks & Dietterich, 2019). We refer readers to the corresponding papers for more details about each dataset. In our main experiments detailed in Section 5.2.1, we did not apply any data augmentations.

**Network.** We conducted our experiments using R20-FRN-Swish, as HMC checkpoints provided by (Izmailov et al., 2021) were publicly available. The model is a modified version of the 20-layer residual network with projection shortcuts (He et al., 2016), incorporating Filter Response Normalization (FRN; Singh & Krishnan, 2020) as the normalization layer and Swish (Hendrycks & Gimpel, 2016; Elfwing et al., 2018; Ramachandran et al., 2017) as the activation function. Substituting Batch Normalization (BN; Ioffe & Szegedy, 2015) with FRN removes the mini-batch dependencies between training examples, while using Swish results in a smoother posterior surface, facilitating a clearer Bayesian interpretation (Wenzel et al., 2020; Izmailov et al., 2021).

**Running from scratch.** In the first setting of the CIFAR experiments, SGMCMC is executed from random initialization to collect posterior samples in a 'from scratch' manner. This represents the most basic setup, requiring SGMCMC methods to quickly reach low posterior energy regions while gathering functionally diverse posterior samples. Starting from the He normal initialization (He et al., 2015), SGMCMC methods were allocated 5,000 steps per sampling cycle (approximately 31 epochs) to generate a total of 100 samples. Table 11 provides detailed hyperparameters.

---

**Algorithm 3:** Practical implementation of SGHMC

**Require:** constant friction value $\gamma$ and the hyperparameters mentioned above.
**Ensure:** a set of posterior samples $\Theta \leftarrow \{\}$.

Initialize position $\boldsymbol{\theta}_{0,T}$ from scratch or pre-set values.
Initialize $\boldsymbol{r}_{0,T} \leftarrow \mathbf{0}$.
**for** $m = 1, 2, \ldots, M$ **do**
    Initialize $(\boldsymbol{\theta}_{m,0}, \boldsymbol{r}_{m,0}) \leftarrow (\boldsymbol{\theta}_{m-1,T}, \boldsymbol{\theta}_{m-1,T})$.
    **for** $t = 1, 2, \ldots, T$ **do**
        $\boldsymbol{z}_{m,t} \sim \mathcal{N}(\mathbf{0}, \mathbf{I})$.
        $\boldsymbol{g}_{m,t} = \nabla_{\boldsymbol{\theta}} \tilde{U}(\boldsymbol{\theta}; \mathcal{B}_{m,t})|_{\boldsymbol{\theta}=\boldsymbol{\theta}_{m,t-1}}$.
        $\boldsymbol{r}_{m,t} = (1 - \gamma\epsilon_t)\boldsymbol{r}_{m,t-1} + \epsilon_t\boldsymbol{g}_{m,t} + \sqrt{2\gamma\epsilon_t}\boldsymbol{z}_{m,t}$.
        $\boldsymbol{\theta}_{m,t} = \boldsymbol{\theta}_{m,t-1} - \epsilon_m\boldsymbol{r}_{m,t}$.
    **end**
    Collect $\Theta \leftarrow \Theta \cup \{\boldsymbol{\theta}_{m,T}\}$.
**end**
**return** $\Theta$

---

**Algorithm 4:** Practical implementation of SGNHT

**Require:** initial thermostat value $\xi$ and the hyperparameters mentioned above.
**Ensure:** a set of posterior samples $\Theta \leftarrow \{\}$.

Initialize position $\boldsymbol{\theta}_{0,T}$ from scratch or pre-set values.
Initialize $\boldsymbol{r}_{0,T} \leftarrow \mathbf{0}$ and $\xi_{0,T} \leftarrow \xi$.
**for** $m = 1, 2, \ldots, M$ **do**
    Initialize $(\boldsymbol{\theta}_{m,0}, \boldsymbol{r}_{m,0}, \xi_{m,0}) \leftarrow (\boldsymbol{\theta}_{m-1,T}, \boldsymbol{\theta}_{m-1,T}, \xi_{m-1,T})$.
    **for** $t = 1, 2, \ldots, T$ **do**
        $\boldsymbol{z}_{m,t} \sim \mathcal{N}(\mathbf{0}, \mathbf{I})$.
        $\boldsymbol{g}_{m,t} = \nabla_{\boldsymbol{\theta}} \tilde{U}(\boldsymbol{\theta}; \mathcal{B}_{m,t})|_{\boldsymbol{\theta}=\boldsymbol{\theta}_{m,t-1}}$.
        $\boldsymbol{r}_{m,t} = (1 - \xi_{m,t-1}\epsilon_t)\boldsymbol{r}_{m,t-1} + \epsilon_t\boldsymbol{g}_{m,t} + \sqrt{2\xi\epsilon_t}\boldsymbol{z}_{m,t}$.
        $\boldsymbol{\theta}_{m,t} = \boldsymbol{\theta}_{m,t-1} - \epsilon_m\boldsymbol{r}_{m,t}$.
        $\xi_t = \xi_{t-1} + \epsilon_t(\frac{\boldsymbol{r}_{t-1}^{\top}\boldsymbol{r}_{t-1}}{n} - 1)$, where $n$ is the dimension of $\boldsymbol{r}_{t-1}$.
    **end**
    Collect $\Theta \leftarrow \Theta \cup \{\boldsymbol{\theta}_{m,T}\}$.
**end**
**return** $\Theta$

---

**Running from HMC burn-in.** The second setting of the CIFAR experiments involves running SGMCMC from HMC burn-in initialization to analyze the dynamics of SGMCMC methods in comparison with the gold-standard HMC. Specifically, we adopted the 50th HMC checkpoint provided by Izmailov et al. (2021), as they designated 50 as the burn-in iteration. To minimize mini-batch noise as much as possible within our computational constraints, a large mini-batch size of 4,096 was employed, aligning with HMC's use of full data to compute gradients. Consequently, using the 50th HMC sample as the initial position, the both SGHMC and PX-SGHMC methods were allocated 70,248 steps per sampling cycle (approximately 7025 epochs), matching the 70,248 leapfrog steps of HMC, to generate a total of 10 samples. We also use the constant step size of $\epsilon_t = 10^{-5}$ and prior variance of $\sigma^2 = 0.2$, in line with HMC.

### D.3 IMAGE CLASSIFICATION WITH DATA AUGMENTATION

**Dataset.** For CIFAR-100, we used 40,960 training examples and 9,040 validation examples, consistent with CIFAR-10. For TinyImageNet, we employed 81,920 training examples and 18,080 validation examples, resizing images from $64 \times 64 \times 3$ to $32 \times 32 \times 3$. In Section 5.2.2, we employed

**Table 11: Hyperparameters for CIFAR.** It summarizes the hyperparameters for each method used in our main evaluation results on the CIFAR experiments (i.e., Tables 1 and 2). If a hyperaparameter was manually set without tuning, it is indicated with a dash in the 'Search Space' column.

| Method | Hyperparameter | Value | Search Space | Notation |
|---|---|---|---|---|
| SGLD | Peak Step Size | $1 \times 10^{-5}$ | $\{3 \times 10^{-k}, 1 \times 10^{-k}\}_{k=4}^{6}$ | $\epsilon_0$ |
| | Prior Variance | 0.2 | $\{0.5, 0.2, 0.1, 0.05, 0.02, 0.01\}$ | $\sigma^2$ |
| pSGLD | Peak Step Size | $3 \times 10^{-4}$ | $\{3 \times 10^{-k}, 1 \times 10^{-k}\}_{k=4}^{6}$ | $\epsilon_0$ |
| | Prior Variance | 0.2 | $\{0.5, 0.2, 0.1, 0.05, 0.02, 0.01\}$ | $\sigma^2$ |
| | Smoothing Factor | 0.99 | $\{0.9, 0.99, 0.999\}$ | $\beta$ |
| SGHMC | Friction | 100 | $\{1, 10, 100, 1000\}$ | $\gamma$ |
| | Peak Step Size | $3 \times 10^{-4}$ | $\{3 \times 10^{-k}, 1 \times 10^{-k}\}_{k=3}^{5}$ | $\epsilon_0$ |
| | Prior Variance | 0.05 | $\{0.5, 0.2, 0.1, 0.05, 0.02, 0.01\}$ | $\sigma^2$ |
| PX-SGHMC | Friction for $\mathbf{V}$ | 100 | $\{1, 10, 100, 1000\}$ | $\gamma$ |
| | Friction for $\mathbf{P}$, $\mathbf{Q}$ | 1 | - | $\gamma$ |
| | Peak Step Size | $1 \times 10^{-4}$ | $\{3 \times 10^{-k}, 1 \times 10^{-k}\}_{k=3}^{4}$ | $\epsilon_0$ |
| | Prior Variance | 0.02 | $\{0.5, 0.2, 0.1, 0.05, 0.02, 0.01\}$ | $\sigma^2$ |
| Shared | Batch Size | 256 | - | $|\mathcal{B}|$ |
| | Step Size Schedule | Cyclical | - | $\epsilon_t$ |
| | Total Updates | $5000 \times 100$ | - | $T$ |
| | Total Samples | 100 | - | $M$ |

a standard data augmentation policy that includes random cropping of 32 pixels with a padding of 4 pixels and random horizontal flipping.

**Network.** We employ R20-FRN-Swish, consistent with the CIFAR-10 experiments.

**Running from scratch.** We utilize the same hyperparameters as in the CIFAR-10 experiments.

## D.4 DATASET DETAILS

Fig. 10 visualizes example images from datasets we considered:

- CIFAR-10 (unknown license); https://www.cs.toronto.edu/ kriz/cifar.html
- CIFAR-10.1 under the MIT license; https://github.com/modestyachts/CIFAR-10.1
- CIFAR-10.2 (unknown license); https://github.com/modestyachts/cifar-10.2
- STL (unknown license); https://cs.stanford.edu/ acoates/stl10/
- SVHN (unknown license); https://github.com/facebookresearch/odin
- LSUN (unknown license); https://github.com/facebookresearch/odin
- CIFAR-100 (unknown license); https://www.cs.toronto.edu/ kriz/cifar.html
- TinyImageNet (unknown license); https://www.kaggle.com/c/tiny-imagenet

## E CONCEPTUAL ILLUSTRATION FOR EFFECT OF PARAMETER EXPANSION

The main motivation for our method is the well-known effects in deep linear neural networks, which can be interpreted as an implicit acceleration of training induced by gradient updates in such networks (Arora et al., 2018). We conceptually illustrated this in Fig. 11.

At a high level, the preconditioning matrix can be understood to evoke an effect akin to adaptive step size scaling, which varies across different components of the parameters. This contrasts with simply increasing the step size, which scales up updates along all axes of parameters by the same factor. Our parameter expansion scales the update along each eigenvector of the preconditioning matrix proportionally to its eigenvalue. This not only amplifies the gradient but also changes its direction.

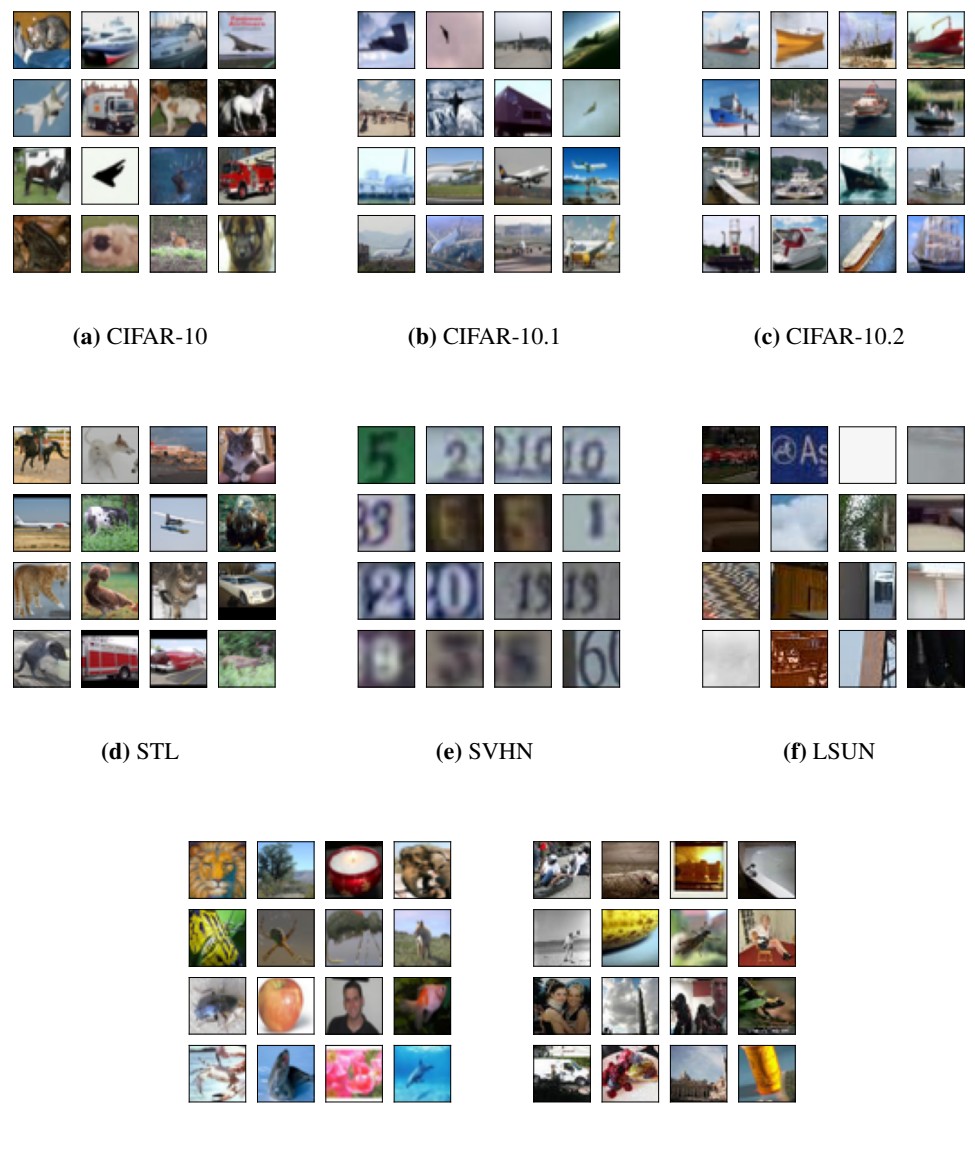

**(a)** CIFAR-10       **(b)** CIFAR-10.1       **(c)** CIFAR-10.2

**(d)** STL       **(e)** SVHN       **(f)** LSUN

**(g)** CIFAR-100       **(h)** TinyImageNet

Figure 10: **Sample images from datasets.** It shows the first 16 test images from CIFAR-like datasets, including (a) CIFAR-10, (b) CIFAR-10.1, (c) CIFAR-10.2, and (d) STL, as well as out-of-distribution datasets such as (e) SVHN and (f) LSUN. Additionally, (g) CIFAR-100 and (h) Tiny-ImageNet datasets are utilized in experiments involving data augmentation.

The eigenvalues of the preconditioner can be mathematically described by the singular values of the merged weight matrix under certain assumptions, including the initialization of the weight matrices as described in Claim 1 of Arora et al. (2018).

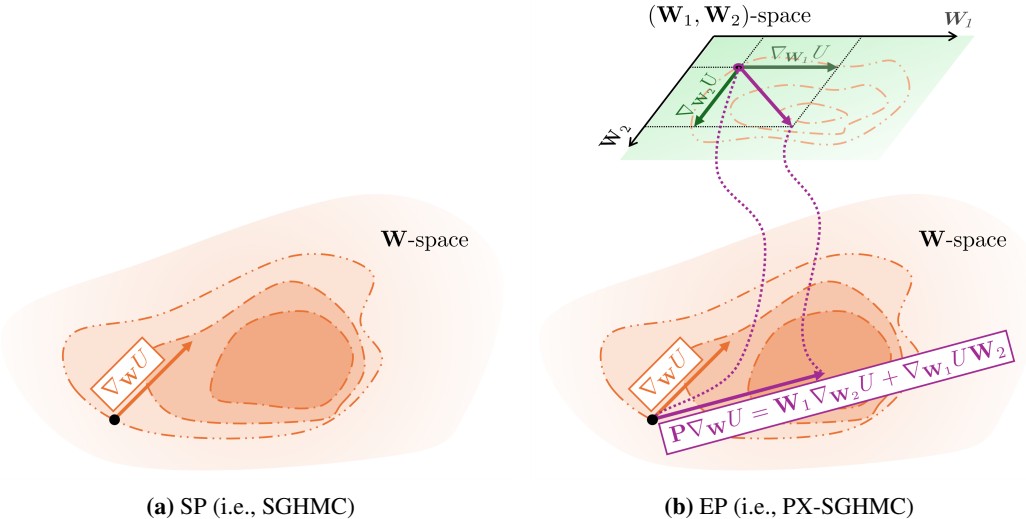

**(a)** SP (i.e., SGHMC)  **(b)** EP (i.e., PX-SGHMC)

**Figure 11: Conceptual illustration comparing SP and EP.** A simplified illustration of the update rules for standard parameterization (SP) and expanded parameterization (EP). The gradient computed in the expanded $(\mathbf{W}_1, \mathbf{W}_2)$-space acts as a preconditioned gradient $\mathbf{P}\nabla_{\mathbf{W}}U$, where $\mathbf{P}$ represents a preconditioner defined in Lemma 3.1, in the original $\mathbf{W}$-space. Compared to the original gradient $\nabla_{\mathbf{W}}U$, the preconditioned gradient is expected to enable more efficient updates by better aligning with the geometry of the parameter space.

