# OpenReview forum: "Parameter Expanded Stochastic Gradient Markov Chain Monte Carlo"
_ICLR.cc/2025/Conference — ICLR 2025 Poster_

### Official Review · Reviewer_Sd8N · 2024-11-03

**Soundness:** 3
**Presentation:** 3
**Contribution:** 2
**Rating:** 6
**Confidence:** 2

**Summary:**

In this paper, the authors studied sampling strategy used in Bayesian Neural Networks (BNNs), and proposed a new approach to enhance sampling diversity in stochastic gradient Markov chain Monte Carlo (SGMCMC). which is one of the most commonly used sampling methods in BNNs. Their proposed approach , known as PX-SGMCMC, can better explore the target parameter space by decomposing the weight matrices int o a produce of matrices. While doing this decomposition, their approach does not increase sampling cost compared against the original SGMCMC. They evaluated PX-SGMCMC on image data, and show that  PX-SGMCMC can indeed achieve more diverse samples, leading to outperformance in uncertainty characterization and robustness.

**Strengths:**

In general, this paper is well-structured and the writing is in good quality. It's great that the authors provided detailed mah derivations for their approach, and to my best knowledge, the proof is sound.
In term of originality, because sample diversity can significantly influence the methods such as SGMCMC, I believe the idea from this paper is well motivated. And the authors provided comprehensive evaluations on multiple image datasets and it is impressive that the PX-SGMCMC can achieve better sample diversity.

**Weaknesses:**

The main concern is how this work is distinguished from the prior work on SGMCMC. Because there are many prior methods (such as the cited work in the Related Work section), I would hope to better understand their unique contributions to this topic.

**Questions:**

1. How to interpret the different colors shown as the distribution modes? It shows a color scale but I don't know what the values indicate.
2. What do the authors think are the possible ways of reducing the computational costs for PX-SGMCMC? The answer does not have to be concrete, it just will be interesting to know conceptually how to optimize the cost in this case.

---

> ### Author Response · Authors · 2024-11-19
>
> Thank you for your positive assessment on our work! We are pleased that you recognized both the soundness of our mathematical derivations and the comprehensiveness of our experiments. We hope the responses below clarify any remaining concerns, and please let us know if there are any additional points to address. We sincerely appreciate your valuable comments.
>
> > How to interpret the different colors shown as the distribution modes? It shows a color scale but I don't know what the values indicate.
>
> The color map represents the negative log probability, with brighter colors indicating lower negative log probability, corresponding to the modes of the distribution. We will clarify the meaning of the contour plot in a future revision.
>
> > The main concern is how this work is distinguished from the prior work on SGMCMC. Because there are many prior methods (such as the cited work in the Related Work section), I would hope to better understand their unique contributions to this topic.
>
> We begin by introducing two main directions in the existing body of work on SGMCMC: (a) the meta-learning approach and (b) the cyclical step size scheduling strategy. Below, we briefly describe these approaches and highlight how our method differs from and complements them.
>
> **(a) Meta-learning Approaches.** After the seminal work of Ma et al. (2015) provided a comprehensive framework for generalizing SGMCMC methods, Gong et al. (2019) attempted to meta-learn the diffusion (D) and curl (Q) matrices, which is typically predefined for each variant of SGMCMC. They introduce additional neural networks estimating Q and D, and define objective functions that lead Q and D to promote the exploration of SGMCMC. After that, due to several practical challenges (as noted by Kim et al. (2024)), Kim et al. (2024) further presented an alternative approach that meta-learns the formulation of the momentum update defined in SGHMC instead. The core idea behind these meta-learning approaches is to enhance exploration in posterior sampling within the SGMCMC framework, which traditionally depends on explicitly defined dynamics. However, due to the added training for meta-learning, they require a huge extra cost in the sampling process.
>
> **(b) Cyclical Step Size Schedule.** Zhang et al. (2020) presented a cyclical step size scheduling in SGMCMC, similar to the cyclical learning rate scheduling used for training deep neural networks (Smith, 2017; Loshchilov and Hutter, 2017). Traditional SGMCMC decays its step size continuously from the initial value, while cyclical SGMCMC periodically resets it to the initial value to recover exploration ability achievable from the larger step size, which can help escape modes of the target distribution despite increasing discretization error. Due to its simplicity and strong performance compared to previous methods, we also used this cyclical scheduling in our experiments including baseline methods.
>
> **Our Contributions.** Ultimately, existing methods for improving SGMCMC aim to enhance sample diversity in the posterior sampling process; meta-learning approaches use meta-learned components to refine the sampling dynamics for better exploration, while the cyclical step size schedule, though simple, effectively promotes exploration. To the best of our knowledge, our approach is the first to improve SGMCMC exploration by simply modifying the parameterization. While our expanded parameterization has been analyzed from an optimization perspective in the context of deep linear neural networks, it has not been explored in the context of posterior sampling within the SGMCMC framework. Our work makes a contribution by comprehensively addressing both the theoretical and experimental aspects of posterior sampling, presenting a new direction for enhancing sample diversity in SGMCMC through simple changes in parameterization. Importantly, our approach is orthogonal and complementary to existing meta-learning and step size scheduling methods, which means that the improvements from each strategy can be combined effectively.

---

> ### Author Response · Authors · 2024-11-19
>
> > What do the authors think are the possible ways of reducing the computational costs for PX-SGMCMC? The answer does not have to be concrete, it just will be interesting to know conceptually how to optimize the cost in this case.
>
> As acknowledged by both us and other reviewers, the cost increases linearly with depth in fully-connected neural networks, which is a main weakness of our method. However, for specialized architectures like convolutional layers, as described in Section 3.3, the relative size of the expanded matrices can be significantly reduced, as they depend on the number of input/output channels in the convolutional layers. Additionally, the memory overhead occurs only during the sampling phase, and the inference cost remains unchanged, since the expanded weights can be combined into a single weight matrix through matrix multiplication.
>
> During the rebuttal period, we analyzed experimentally our approach from the perspective of the wall-clock time and burn-in in practical setups, as suggested by Reviewer 53Wx. We also implemented a low-rank variant of our approach suggested by Reviewer Awai which reduces memory overhead. To summarize: 1) the additional memory and time overhead introduced by PX-SGHMC is not too huge, with only x1.01 in GPU memory usage and x1.05 in wall-clock time in practical implementation, 2) introducing EP is more beneficial than introducing burn-in into SGHMC, and 3) the low-rank variant allows for a more memory-efficient implementation of EP. We encourage you to refer to those responses for further details.
>
> ---
> Gong et al. (2019), Meta-Learning for Stochastic Gradient MCMC.
> Kim et al. (2024), Learning to Explore for Stochastic Gradient MCMC.
> Loshchilov and Hutter (2017), SGDR: Stochastic Gradient Descent with Warm Restarts.
> Ma et al. (2015), A Complete Recipe for Stochastic Gradient MCMC.
> Smith (2017), Cyclical Learning Rates for Training Neural Networks.
> Zhang et al. (2020), Cyclical Stochastic Gradient MCMC for Bayesian Deep Learning.

---

> > ### Comment · Reviewer_Sd8N · 2024-11-26
> >
> > I would like to thank the authors for their explanations and I believe my questions have been addressed properly. I decide to keep my original score.

---

> > > ### Author Response · Authors · 2024-11-27
> > >
> > > Thank you for taking the time to review our explanations. We are pleased to hear that your questions have been addressed properly.

---

### Official Review · Reviewer_53Wx · 2024-11-03

**Soundness:** 3
**Presentation:** 3
**Contribution:** 3
**Rating:** 8
**Confidence:** 4

**Summary:**

The authors propose an approach to enhance sample diversity in SGMCMC for Bayesian neural networks.  Their approach is based on decomposing the weight matrices of each BNN layer into a product matrices.  They provide theoretical justification for this approach by showing that the euclidean distance between consecutive samples depends on the number of matrices used in the product construction.  They validate their approach on toy data and several classification tasks.  their results show empirical correspondence to the derived theorem, and show their method helps to provide robustness to distribution shifts, can enhance OOD detection and increase sample diversity.

**Strengths:**

the paper is clearly written and easy to read.  it is impressive that the authors provide theoretical justification for their proposed parameter expanded SGMCMC by showing that the expected euclidean distance between parameters is bounded above by a quantity depending on the depth of the weight matrix parameterization.

the authors provide a few variants of EP for different architectures showing that it can be extended beyond linear layers in theory and in practice (as justified by their latter experiments).  they provide extensive experimental/empirical results that show their proposed parameterization can enhance sampling quality in BNNs.  they examine several properties like robustness to distribution shift, out of distribution detection, and show their method also provides improvements even when data augmentation is used.  the number of experiments and their coverage of standard questions readers/researchers might be interested in is both comprehensive and extensive.

**Weaknesses:**

in section 3, it might be useful to hold the reader’s hand more and expand on lines 168-174 in a way that makes statements about the preconditioning aspect more precise and perhaps easier to visually parse.  for example i am not sure what was meant by the preconditioning producing `extraordinary directions of gradient steps.’


the authors remark about increased parameter counts during training, but it might be helpful to see wallclock times as well as burn in times.  maybe it is possible that drawing single samples is slower, but the enhanced training dynamics lead to a smaller burn in period, which could be another plus for the parameter expanded neural network layers if that difference were sufficiently large.

**Questions:**

I am confused about the variables introduced in Eq.11 — in particular, it could be helpful to see how they relate to $r$ for the langevin dynamics of the classic unaltered network.

while fig.2 shows that the maximum singular value increases while the minimum singular value decreases, how does the condition number change?

---

> ### Author Response · Authors · 2024-11-19
>
> Thank you for your thoughtful and positive feedback on our work. We’re delighted that you recognized both our theoretical contributions and the extensiveness of our experiments. We hope the responses below resolve any remaining concerns, but please feel free to let us know if there are additional issues. We truly appreciate your valuable comments!
>
> > in section 3, it might be useful to hold the reader’s hand more and expand on lines 168-174 in a way that makes statements about the preconditioning aspect more precise and perhaps easier to visually parse. for example i am not sure what was meant by the preconditioning producing `extraordinary directions of gradient steps.’
>
> Thank you for pointing out this ambiguous expression. By “extraordinary directions of gradient steps,” we meant that the effect of preconditioning is similar to increasing the step size, but not uniformly across all components. The preconditioners introduced by our parameter expansion apply different scaling factors to different subspaces, hence changing the direction of the gradient update. To better support the reader’s understanding, we have included a conceptual visualization comparing the expected direction of PX-SGMCMC to standard SGMCMC, as shown in Figure 11 in Appendix E of the revised PDF.
>
> > the authors remark about increased parameter counts during training, but it might be helpful to see wallclock times as well as burn in times. maybe it is possible that drawing single samples is slower, but the enhanced training dynamics lead to a smaller burn in period, which could be another plus for the parameter expanded neural network layers if that difference were sufficiently large.
>
> We truly appreciate the new perspective you've provided on wall-clock time and the burn-in period in validating the effectiveness of our method. These additional ablation studies can be found in Appendices B.6 and B.7 of the revised manuscript, and below is a brief summary.
>
> Regarding wall-clock time, we have provided system logs comparing PX-SGHMC with SGHMC in Table 7 of the revised manuscript. The logs show that PX-SGHMC demonstrates no significant difference in sampling speed or memory consumption when compared to SGHMC in practice. Of course, these outcomes are primarily due to the use of the relatively small R20 model; if much larger models (e.g., modern transformer architectures) were employed, practical GPU memory allocation (i.e., "Space (in practice)") would be more strongly influenced by the number of parameters (i.e., "Space (in theory)"). In such cases, we believe that the memory overhead reduction strategy via low-rank constraints, which we discussed with Reviewers Awai and Sd8N, would be particularly beneficial.
>
> Regarding burn-in, none of the SGMCMC methods in our experiments includes a separate burn-in phase; in other words, even samples from the first cycle are used for BMA computation. Inspired by your comments, we conducted an additional ablation study on burn-in. Specifically, this experiment is similar to the one performed by Izmailov et al. (2021) for HMC. In Figure 9 of the revised PDF, the x-axis represents the number of burn-in samples, i.e., the length of the burn-in period, while the y-axis shows the individual performance of the 50 samples (in average) obtained after the burn-in period (denoted as "IND (1)") and the ensemble performance (denoted as "BMA (50)"). For reference, we also plotted the previously reported performance without a burn-in period (denoted as "BMA (100)"). It clearly demonstrates that the "enhanced training dynamics" introduced by our EP enables higher BMA performance with a shorter burn-in period. In other words, when collecting the same 50 samples, SGHMC requires a much longer burn-in period to achieve performance comparable to that of PX-SGHMC.
>
> > I am confused about the variables introduced in Eq.11 — in particular, it could be helpful to see how they relate to $r$ for the langevin dynamics of the classic unaltered network.
>
> You can consider that the standard SGMCMC (SP) only has $(\mathbf{S}^{(l)})_{l=1}^L$ from Eq. (11). The other momentum variables such as $\mathbf{R}$ and $\mathbf{T}$ are introduced by the expanded parameters in the EP setting. For example, $\mathbf{R}$ is required to be a momentum for the weight $\mathbf{P}$ and likewise $\mathbf{T}$ corresponds to $\mathbf{Q}$. We will clarify those in the revised version of the paper.
>
> > while fig.2 shows that the maximum singular value increases while the minimum singular value decreases, how does the condition number change?
>
> As you predicted based on the previously presented largest and smallest singular value plots, the condition number also increases with depth, leading to a larger overall scale across layers. We have updated Figure 2 to include the condition number plot, and we encourage you to refer to it.
>
> ---
> Izmailov et al. (2021), What Are Bayesian Neural Network Posteriors Really Like?

---

### Official Review · Reviewer_Awai · 2024-11-03

**Soundness:** 4
**Presentation:** 4
**Contribution:** 3
**Rating:** 8
**Confidence:** 3

**Summary:**

In this paper, the authors address the issue of limited sample diversity when using Stochastic Gradient Markov Chain Monte Carlo (SGMCMC) for posterior sampling in Bayseian Neural Networks (BNNs). They propose a novel expanded parametrization (EP) of neural networks that relies on a decomposition of the network weights for producing diverse sample weights. They provide theoretical and empirical validation of the proposed approach.

**Strengths:**

The paper is well structured and the writing is easy to follow. The approach is fairly practical, easy to adopt for general neural network architectures and circumvents the need for high compute resources associated with running multiple chains. Moreover, the theoretical analysis on the gradient flow of EP and the corresponding bound on distance between consecutive samples provides interesting insights on the relation between the induced network “depth”, maximum singular value of the matrices and exploration during sampling. The empirical evaluations clearly demonstrate the strength of the proposed approach for robustness to distribution shifts, ood performance and further validate the theoretical bound.

**Weaknesses:**

* One of the weaknesses, as noted in the paper, is the additional overhead introduced by the EP. Since the sample diversity depends on the depth of the EP, this could potentially be prohibitive when dealing with large networks.
* The authors note that one of the reasons why less principled deep ensembles perform better than SGMCMC is better sample diversity. It would be helpful to see how the proposed approach compares to deep ensembles.

**Questions:**

* Have the authors tried comparing the proposed method to one of the meta-learning approaches proposed for increasing sample diversity (Gong et al., 2019; Kim et al., 2024)?
* Do the authors have any ideas on modifying the EP design to address computational overhead? For instance, I’m curious to know if introducing low-rank constraints in these matrices significantly impact the results?

---

> ### Author Response · Authors · 2024-11-19
>
> We are happy that you enjoyed both the theoretical analysis and the empirical evaluation we presented. We hope the responses below address any remaining concerns. Please let us know If there are any further issues. Thank you again for your valuable comments!
>
> > One of the weaknesses, as noted in the paper, is the additional overhead introduced by the EP. Since the sample diversity depends on the depth of the EP, this could potentially be prohibitive when dealing with large networks.
> >
> > Do the authors have any ideas on modifying the EP design to address computational overhead? For instance, I’m curious to know if introducing low-rank constraints in these matrices significantly impact the results?
>
> As discussed in Section 3.3, the expanded matrices for convolutional layers depend on the input and output channels. Thus, in convolutional neural networks, our proposed EP does not cause the two- to three-fold increase in parameters observed in fully connected networks. For example, in our CIFAR-10 experiments, PX-SGHMC uses approximately 1.4 times more parameters during sampling compared to SGHMC (274,042 for SGHMC and 383,098 for PX-SGHMC; importantly, these additional parameters are only used during sampling and can be fused when saving each sample). Nonetheless, we acknowledge that this increase in parameters is a clear weakness of the EP method. Adopting a low-rank constraint, as suggested, offers a promising solution to reduce the parameter count and mitigate this issue.
>
> In this regard, we conducted additional experiments using a low-rank plus diagonal approach to construct the expanded matrices. Specifically, we compose a new expanded matrix $\mathbf{P} = \mathbf{D} + \mathbf{L}_1^\top\mathbf{L}_2$ for a diagonal matrix $\mathbf{D} \in \mathbb{R}^{p \times p}$ and the low-rank matrices $\mathbf{L}_1, \mathbf{L}_2 \in \mathbb{R}^{r \times p}$ with $r < p$, which reduces the memory from $O(p^2)$ to $O(p + 2pr)$. In the following table, we set $r$ to $p / 8$ and $p / 4$ for the $c=d=1$ setup, resulting in 303,610 and 330,874 parameters during the sampling procedure, respectively. Even with the expanded matrices set to the low-rank plus diagonal form, PX-SGHMC continues to outperform SGHMC, highlighting a clear direction for effectively addressing the increased parameter count of our method. We appreciate your constructive comments, which offer valuable insights into ways to mitigate the overhead introduced by the EP. These results have been included in Appendix B.6 of the revised PDF. Thank you for your thoughtful feedback!
>
> | Label               | # Params (sampling) | # Params (inference) | ERR   | NLL   | AMB   |
> | :-                  | :-                  | :-                   | :-    | :-    | :-    |
> | SGHMC               | 274,042 (x1.00)     | 274,042 (x1.00)      | 0.131 | 0.421 | 0.183 |
> | PX-SGHMC (low-rank) | 303,610 (x1.11)     | 274,042 (x1.00)      | 0.126 | 0.401 | 0.210 |
> | PX-SGHMC (low-rank) | 330,874 (x1.21)     | 274,042 (x1.00)      | **0.122** | **0.385** | 0.215 |
> | PX-SGHMC            | 383,098 (x1.40)     | 274,042 (x1.00)      | 0.123 | 0.396 | **0.242** |

---

> ### Author Response · Authors · 2024-11-19
>
> > The authors note that one of the reasons why less principled deep ensembles perform better than SGMCMC is better sample diversity. It would be helpful to see how the proposed approach compares to deep ensembles.
>
> As you might know, deep ensemble (DE) typically outperforms Bayesian methods (Ashukha et al., 2020; Ovadia et al., 2019), including the SGMCMC methods we focus on—particularly when a single chain is used in the SGMCMC setup (Izmailov et al., 2021). Wilson and Izmailov (2021) argued that the DE framework, which collects diverse low-loss solutions from multiple random initializations at a substantial training cost, offers a strong approximation for the numerical integration of the predictive BMA. While our manuscript has already presented a comparative study using HMC checkpoints from Izmailov et al. (2021), which yield more diverse posterior samples at a higher computational cost than DE, additional results comparing HMC, DE, SGHMC, and PX-SGHMC in a from-scratch setup would provide valuable insights for readers and future researchers alike.
>
> The following table provides a summary: (a) “# Samples” indicates “{number of chains}x{number of samples per chain},” representing the total samples used for ensembling. (b) “# Epochs” refers to the total training epochs for constructing each ensemble. (c) Lower “ERR” and “NLL” values correspond to better predictive performance. (d) Higher “AMB” values indicate greater predictive diversity. Here, DE with # Epochs of 31,250 consists of 100 training runs, each with 50,000 iterations (equivalent to approximately 312.5 training epochs in our experiments). As previously observed, DE achieves predictive performance (ERR and NLL) on par with HMC while maintaining high predictive diversity (AMB). When treating the predictive diversity of HMC and DE as a reference point, our PX-SGHMC exhibits substantially higher predictive diversity (AMB) than SGHMC in a single-chain setup, along with improved predictive performance in terms of ERR and NLL.
>
> | Method   | # Samples | # Epochs   | ERR  | NLL  | AMB  |
> | :-       | :-        | :-         | :-   | :-   | :-   |
> | HMC      | 3x240     | 61,115,760 | .093 | .307 | .513 |
> | HMC      | 1x240     | 20,371,920 | .107 | .326 | .471 |
> | DE       | 100x1     | 31,250     | .107 | .316 | .314 |
> ||
> | PX-SGHMC | 1x100     | 3,125      | .121 | .388 | .242 |
> | SGHMC    | 1x100     | 3,125      | .135 | .422 | .186 |
>
> > Have the authors tried comparing the proposed method to one of the meta-learning approaches proposed for increasing sample diversity (Gong et al., 2019; Kim et al., 2024)?
>
> We are pleased to present comparative results with L2E (Kim et al., 2024) using their publicly available official implementation. Since they also adopted the experimental setup of Izmailov et al. (2021) using the R20-FRN-Swish architecture, the results are largely comparable when key experimental configurations—such as data augmentation, the number of steps per cycle, and the number of posterior samples—are aligned. Using the same setup as in Table 3 of our manuscript (i.e., with data augmentation, 5000 steps per cycle, and 100 posterior samples), we ran the L2E algorithm and obtained the following results.
>
> Notably, our PX-SGHMC, which applies a vanilla SGHMC sampler to the expanded parameterization, yields competitive results compared to L2E, despite L2E relying on a more resource-intensive meta-learned sampler. Furthermore, it is important to emphasize that our method enhances the parameterization, while L2E focuses on enhancing the sampler; it makes the orthogonal and complementary improvements (i.e., they can be combined). These comparative results with the L2E approach are presented in Appendix B.8 of the revised PDF, further strengthening our paper. We sincerely appreciate the constructive feedback!
>
> | Method   | ERR@C10  | ERR@C100 | ERR@TIN  | NLL@C10  | NLL@C100 | NLL@TIN  |
> | :-       | :-        | :-        | :-        | :-        | :-        | :-        |
> | SGHMC    | 0.071    | 0.319    | 0.538    | 0.223    | 1.138    | 2.251    |
> | PX-SGHMC | **0.069**    | 0.290    | 0.498    | **0.217**    | 1.030    | 2.089    |
> | L2E      | 0.071    | **0.268**    | **0.488**    | 0.232    | **0.980**    | **2.062**    |
>
> ---
> Ashukha et al. (2020), Pitfalls of In-Domain Uncertainty Estimation and Ensembling in Deep Learning.
> Izmailov et al. (2021), What Are Bayesian Neural Network Posteriors Really Like?
> Kim et al. (2024), Learning to Explore for Stochastic Gradient MCMC.
> Ovadia et al. (2019), Can You Trust Your Model’s Uncertainty? Evaluating Predictive Uncertainty Under Dataset Shift.
> Wilson and Izmailov (2021), Deep Ensembles as Approximate Bayesian Inference.

---

> > ### Comment · Reviewer_Awai · 2024-11-25
> >
> > I would like to thank the authors for their detailed response addressing my concerns. I appreciate the work put in by the authors to include the additional comparison, and I'm glad to know that the low-rank plus diagonal EP offers strong performance as well. Overall, I think this is a good contribution and I will be maintaining my score.

---

> > > ### Author Response · Authors · 2024-11-27
> > >
> > > Thank you for taking the time and recognizing our efforts. We are glad the performance of the low-rank plus diagonal EP met your expectations.

---

### Official Review · Reviewer_GmGR · 2024-11-09

**Soundness:** 3
**Presentation:** 3
**Contribution:** 3
**Rating:** 6
**Confidence:** 4

**Summary:**

This paper proposes a new SGMCMC method using the idea of parameter expansion to improve the diversity of samples. Specifically, the weight and bias terms in neural networks are reparameterized using a sequence of matrix multiplication. The upper bound on the differences between two consecutive time steps is given. The experiments consider distribution shifts and of-distribution detection on CIFAR datasets.

**Strengths:**

- The paper studies an important problem in SGMCMC, which is to improve sample diversity.
- The method is simple and the algorithm is easy to implement in practice.
- The empirical results show improvement over existing SGMCMC methods.

**Weaknesses:**

- The main concern I have is the motivation of the proposed method. It is unclear why parameter expansion can solve the sample diversity issue. The paper did not explain the intuition and only explain it through Theorem 3.2. However, the theorem did not provide a convincing motivation either.
- In Theorem 3.2, c and d have similar effects as the step size. I believe the values of c and d cannot be too large to achieve good convergence. If so, how is it different from using larger step sizes? Also, does Theorem 3.2 include the standard SGMCMC as a special case? In experiments, the authors use c=0,d=0 to refer to the standard SGMCMC, but the formulation in Eq.9 suggests c,d>=1.
- Another main concern is that the required additional memory and computation will increase significantly. For an m by n W, we now will have c m by m and d n by n matrices, which at least increases O(c+d) costs.
- In experiments, the proposed method did not compare with SGMCMC methods that aim to improve sample diversity, such as cyclical SGMCMC.
- Fig 1 seems to show that EP can find modes but cannot converge to the target distribution, as the bottom left region seems to have more density.
- Is the proposed parameter expansion implemented upon cyclical SGMCMC? This is because results in Fig 3 and 4 suggest so. Also, the authors use the cycle index in several figures. However, it is not explicitly mentioned anywhere in the paper whether PX-SGMCMC is built upon cSGMCMC.
- The empirical performance does not match typical benchmarks. E.g. err 29% on C100 where the neural network with the same architecture typically can achieve <20%.

**Questions:**

NA

---

> ### Author Response · Authors · 2024-11-19
>
> Thank you for your time and effort in reviewing our work! We hope the responses below address any remaining concerns. Please feel free to let us know if there are any further issues. Otherwise, we kindly request that you revise your assessment accordingly. Thank you again!
>
> > The main concern I have is the motivation of the proposed method. It is unclear why parameter expansion can solve the sample diversity issue. The paper did not explain the intuition and only explain it through Theorem 3.2. However, the theorem did not provide a convincing motivation either.
>
> Thank you for the constructive comments. As you mentioned, it is not trivial to develop an intuition about the difference between using a large step size and the implicit step size induced by the preconditioner in Lemma 3.1. The main motivation for our method is the well-known effects in deep linear neural networks, which can be interpreted as an implicit acceleration of training induced by gradient updates in such networks (Arora et al., 2018). We conceptually illustrated this in Figure 11 in Appendix E, shown in the revised PDF.
>
> At a high level, the preconditioning matrix can be understood to evoke an effect akin to adaptive step size scaling, which varies across different components of the parameters. This contrasts with simply increasing the step size, which scales up updates along all axes of parameters by the same factor. Our parameter expansion scales the update along each eigenvector of the preconditioning matrix proportionally to its eigenvalue. This not only amplifies the gradient but also changes its direction. The eigenvalues of the preconditioner can be mathematically described by the singular values of the merged weight matrix under certain assumptions, including the initialization of the weight matrices as described in Claim 1 of Arora et al. (2018).
>
>
> > In Theorem 3.2, c and d have similar effects as the step size. I believe the values of c and d cannot be too large to achieve good convergence. If so, how is it different from using larger step sizes?
>
> As shown in Figure 3, the performance of the BMA (ensemble) improves and then saturates as the depth ($c + d$) increases. Although $c = d = 2$ (making $c + d = 4$) might not seem too large, Eq. (15) in Theorem 3.2 indicates that this choice effectively results in approximately $5M^4$ times longer exploration than standard SGMCMC, where $\epsilon$ is the step size and $M$ is the largest singular value of the weight matrices. In other words, $c + d = 4$ behaves like a step size that is $5M^4$ times larger, which would typically introduce a significant discretization error in SGMCMC if we merely increase the step size. We can also notice that even the largest singular value tends to increase as the depth grows as in the third column in Figure 2.
>
> > Also, does Theorem 3.2 include the standard SGMCMC as a special case? In experiments, the authors use c=0,d=0 to refer to the standard SGMCMC, but the formulation in Eq.9 suggests c,d>=1.
>
> We apologize for the confusion regarding the minimum depth assumed in Theorem 3.2. Theorem 3.2 indeed includes the case where $c = d = 0$, representing standard SGMCMC. Although Eq. (9) is written as if $c, d \geq 1$, both Lemma 3.1 and Theorem 3.2 allow for $c$ and $d$ to be zero. To clarify this and reduce confusion, we have revised the main text to specify $P_{1:0} := I$ and $Q_{1:0} := I$ in Eq. (9).
>
> > Another main concern is that the required additional memory and computation will increase significantly. For an m by n W, we now will have c m by m and d n by n matrices, which at least increases O(c+d) costs.
>
> As acknowledged by both us and other reviewers, the cost increases linearly with depth in fully-connected neural networks, which is a main weakness of our method. However, for specialized architectures like convolutional layers, as described in Section 3.3, the relative size of the expanded matrices can be significantly reduced, as they depend on the number of input/output channels in the convolutional layers. Additionally, the memory overhead occurs only during the sampling phase, and the inference cost remains unchanged, since the expanded weights can be combined into a single weight matrix through matrix multiplication.
>
> During the rebuttal period, we analyzed experimentally our approach from the perspective of the wall-clock time and burn-in in practical setups, as suggested by Reviewer 53Wx. We also implemented a low-rank variant of our approach suggested by Reviewer Awai which reduces memory overhead. To summarize: 1) the additional memory and time overhead introduced by PX-SGHMC is not too huge, with only x1.01 in GPU memory usage and x1.05 in wall-clock time in practical implementation, 2) introducing EP is more beneficial than introducing burn-in into SGHMC, and 3) the low-rank variant allows for a more memory-efficient implementation of EP. We encourage you to refer to those responses for further details.

---

> ### Author Response · Authors · 2024-11-19
>
> > In experiments, the proposed method did not compare with SGMCMC methods that aim to improve sampling diversity, such as cyclical SGMCMC.
> >
> > Is the proposed parameter expansion implemented upon cyclical SGMCMC? This is because results in Fig 3 and 4 suggest so. Also, the authors use the cycle index in several figures. However, it is not explicitly mentioned anywhere in the paper whether PX-SGMCMC is built upon cSGMCMC.
>
> Sorry for the confusion. Currently, the step size schedule is mentioned only in Table 11 (previously Table 7), which specifies that "Cosine" scheduling (equivalent to the cyclical schedule you mentioned, as we described in Appendix C; the revised manuscript now refers to it as “Cyclical” instead of “Cosine”) is used for all SGMCMC methods in the CIFAR experiments. As you pointed out, all methods indeed employ a cyclical step size schedule. We will clarify this in the main text with a statement such as, “Unless otherwise specified, all the SGMCMC methods utilize the cyclical step size schedule.” Thank you for the constructive feedback!
>
> We also present experimental results using a constant step size schedule, highlighting the unique effectiveness of our proposed EP, independent of the choice of step size schedule. The table below summarizes the performance (ERR, NLL; lower is better) and functional diversity (AMB; higher is preferred) of SGHMC and PX-SGHMC under both constant and cyclical step size schedules. Based on "SGHMC w/ constant," the results clearly show that our expanded parameterization (EP) contributes more significantly to performance improvements than the adoption of the cyclical schedule (CS). These findings are also detailed in Appendix B.9 of the revised PDF. We appreciate your constructive feedback!
>
> | Label                | CS | EP | ERR   | NLL   | AMB   |
> | :-                   | :- | :- | :-    | :-    | :-    |
> | SGHMC w/ constant    |    |    | 0.135 | 0.441 | 0.209 |
> | SGHMC w/ cyclical    | ✔  |    | 0.133 | 0.422 | 0.186 |
> | PX-SGHMC w/ constant |    | ✔  | **0.115** | **0.379** | 0.232 |
> | PX-SGHMC w/ cyclical | ✔  | ✔  | 0.121 | 0.388 | **0.242** |
>
> > Fig 1 seems to show that EP can find modes but cannot converge to the target distribution, as the bottom left region seems to have more density.
>
> As you may already know, every MCMC method struggles with sampling from multi-modal distributions, which typically requires numerous parallel chains or an extremely large number of MCMC steps. For instance, Zhang et al. (2020) demonstrated results on a multi-modal distribution using 4 parallel chains, as shown in Figure 2 of their paper. In our demonstration, since SGMCMC begins at the left-bottom-most mode, it predominantly samples from this starting mode for both SP and EP.
>
> > The empirical performance does not match typical benchmarks. E.g. err 29% on C100 where the neural network with the same architecture typically can achieve <20%.
>
> We respectfully disagree with the claim that our empirical results do not align with typical benchmarks. To conduct the comparative experiments described in Section 5.3, we utilized the reference HMC samples provided by Izmailov et al. (2021) and thus employed the R20-FRN-Swish architecture throughout our experiments. Even with data augmentation, achieving a classification error below 20% on CIFAR-100 is highly challenging in this setup; such performance is achievable only with significantly larger models, such as WRN28x10-BN-ReLU, which has 36.5 million parameters—over 100 times more than R20 (cf. google/uncertainty-baselines repo).
>
> As Kim et al. (2024) also report the performance of R20-FRN-Swish on CIFAR-100 with data augmentation, their results serve as the most relevant reference for our empirical performance. In their Table 10, they show that even the deep ensemble achieves a classification error of around 30%.
>
> ---
> Arora et al. (2018), On the Optimization of Deep Networks: Implicit Acceleration by Overparameterization.
> Izmailov et al. (2021), What Are Bayesian Neural Network Posteriors Really Like?
> Kim et al. (2024), Learning to Explore for Stochastic Gradient MCMC.
> Zhang et al. (2020), Cyclical Stochastic Gradient MCMC for Bayesian Deep Learning.

---

> > ### Comment · Reviewer_GmGR · 2024-11-25
> >
> > Thank you for your detailed responses.
> >
> > I found the visualization in Figure 10 very helpful in clarifying the motivation behind your approach. The key benefit of the proposed method seems to be introducing a novel preconditioner. Could you elaborate further on how this preconditioner differs from existing ones in terms of performance and computational costs? What specific advantages does it offer? If the main benefit is the cost, could the cost of existing preconditioners also be reduced using approximations similar to those you used for the proposed method?

---

> > > ### Author Response · Authors · 2024-11-25
> > >
> > > We’re truly glad to hear that the conceptual figure we added was helpful! As you pointed out, our EP indeed functions as a novel preconditioner, and comparing it to existing methods that explicitly introduce preconditioners would certainly be valuable. A key advantage of our proposed EP is that it _implicitly_ introduces a powerful preconditioning effect in the SGMCMC dynamics by simply expanding the parameters, _without requiring the significant computational overhead of explicitly calculating a preconditioner such as a Hessian matrix_.
> > >
> > >
> > > Specifically, the computational cost of existing methods that explicitly introduce preconditioners can also be reduced by adopting approximations, such as diagonal approximations with RMSProp or the diagonal empirical Fisher. While not identical to our low-rank plus diagonal approach, these approximations are commonly used and effective when working with high dimensional target distributions like Bayesian deep neural networks. In our experiments, we demonstrated that both our original method and its cost-efficient variant outperform pSGLD (as shown in Tables 1 and 8), which computes a preconditioner matrix in an RMSProp-like manner (as shown in Algorithm 2), highlighting that our approach surpasses pSGLD, a representative method for explicitly introducing preconditioners.

---

> > > > ### Comment · Reviewer_GmGR · 2024-11-26
> > > >
> > > > Thank you very much for the explanation. I still have some questions about the benefits of the proposed method. To be more concrete,
> > > >
> > > > 1. The claimed benefit of "without requiring the significant computational overhead of explicitly calculating a preconditioner such as a Hessian matrix" seems not true, as the proposed method does not really decrease the cost. I agree that parameter expansion is a new way to introduce preconditioners, but its advantages are unclear to me. What is the reason that it performs better than pSGLD? When should users choose this algorithm over others?
> > > >
> > > > 2. Can you provide explicit cost comparison (theoretically and empirically) for original and approximation versions?

---

> > > > > ### Author Response · Authors · 2024-11-27
> > > > >
> > > > > > 1. The claimed benefit of "without requiring the significant computational overhead of explicitly calculating a preconditioner such as a Hessian matrix" seems not true, as the proposed method does not really decrease the cost. I agree that parameter expansion is a new way to introduce preconditioners, but its advantages are unclear to me. What is the reason that it performs better than pSGLD? When should users choose this algorithm over others?
> > > > >
> > > > > In a nutshell, pSGLD suffers from two major limitations in deep neural network setups in terms of the inaccurate SGMCMC dynamics: *1) approximating the metric using RMSProp preconditioner, which is a highly biased estimator, and 2) omitting the correction term entirely*. Together, these factors result in its relatively poor practical performance, as observed in SGMCMC literature where pSGLD often underperforms SGLD (Kim et al., 2020; Lange et al., 2023), also consistent with our results in Tables 1-3. In contrast, our method introduces the precondition without such a poor approximation of the metric but rather just altering the parameterization of the neural network.
> > > > >
> > > > > More precisely, assuming the SDE dynamics operate on a manifold characterized by a metric $G(\boldsymbol{\theta}) \in \mathbb{R}^{N \times N}$, where $N$ represents the number of parameters, an optimal approach involves introducing metrics that accurately capture the local curvature of the target distribution (e.g., the inverse of the Hessian). Girolami and Calderhead (2011) showed that using the Fisher information matrix as the metric can enhance exploration in small problems where its computation is feasible. Extending this idea, methods like SGRLD (Patterson and Teh, 2013), SGRHMC (Ma et al., 2015), and pSGLD (Li et al., 2016) have employed the Fisher information metric or general Riemannian metrics to precondition SGMCMC dynamics.
> > > > >
> > > > >
> > > > > For deep neural networks, however, 1) computing such metrics in their full form is prohibitively expensive due to high dimensionality of neural network parameters, hence we referred to “the significant computational overhead of explicitly calculating a preconditioner such as a Hessian matrix.” In contrast, our method avoids computing the inverse of the Hessian matrix (which requires over $O(N^3)$, where $N$ is the number of parameters) or the Fisher information matrix (which requires $O(MN^2)$, where $M$ is the number of training data and $N$ is the number of parmaeters); the additional computational cost in our method increases linearly with the number of the expanded parameters (which remains efficiently in $O(N)$).
> > > > >
> > > > >
> > > > > Additionally, incorporating $G$ into diffusion and curl matrices lead to the computation of a correction term $\Gamma$ to ensure valid posterior sampling, involving derivatives with respect to the parameters $W$, which is computationally impractical for the neural network parameters. Thus, SGMCMC researchers typically omit this term in practice, introducing estimation errors (Li et al., 2016; Yu et al., 2023). As a result, pSGLD suffers from two major limitations in deep neural network setups: 1) approximating the metric using RMSProp and 2) omitting the correction term entirely. Together, these factors result in its relatively poor practical performance, as observed in SGMCMC literature where pSGLD often underperforms compared to SGLD (Kim et al., 2020; Lange et al., 2023), consistent with our results in Tables 1-3.
> > > > >
> > > > >
> > > > > As you mentioned, our proposed parameter expansion introduces a novel and practically effective way to incorporate preconditioning effects. Its superior performance, outperforming both pSGLD—the scalable approach for preconditioning SGMCMC dynamics—and the robust baseline of vanilla SGHMC (with standard parameterization), highlights the value of our parameter expansion methodology. Also, our method alters only the parametrization of the target distribution (even for the low-rank variants) and it still follows the same SGMCMC algorithm that estimates a Bayesian neural network with expanded parameters, which does not introduce extra bias on the SGMCMC dynamics.
> > > > >
> > > > > ---
> > > > > Girolami and Calderhead (2011). Riemann manifold Langevin and Hamiltonian Monte Carlo methods.
> > > > > Li et al. (2016). Preconditioned Stochastic Gradient Langevin Dynamics for Deep Neural Networks.
> > > > > Kim et al. (2020). Stochastic Gradient Langevin Dynamics Algorithms with Adaptive Drifts.
> > > > > Yu et al. (2023). Scalable Stochastic Gradient Riemannian Langevin Dynamics in Non-Diagonal Metrics.
> > > > > Lange et al. (2023). Batch Normalization Preconditioning for Stochastic Gradient Langevin Dynamics.

---

> ### Author Response · Authors · 2024-11-27
>
> > 2. Can you provide explicit cost comparison (theoretically and empirically) for original and approximation versions?
>
> It appears that your statement refers to our original version and the low-rank plus diagonal version. Table 8 summarizes the memory cost and performance, and we have measured practical runtimes as 64 sec/cycle and 69 sec/cycle, respectively, which are not significantly higher than the standard SGHMC's 61 sec/cycle. The low-rank plus diagonal version is designed to reduce memory costs, though it incurs a slight increase in runtime due to the calculation of the low-rank and diagonal components.

---

> > ### Comment · Reviewer_GmGR · 2024-11-27
> >
> > Thank you very much for the detailed explanation. My questions have been resolved. I encourage the authors to incorporate some of this explanation into the manuscript to clarify the contribution. I have updated my score.

---

> > > ### Author Response · Authors · 2024-11-28
> > >
> > > We sincerely appreciate the time you took to review our explanations and are glad to hear that your questions have been resolved. We will carefully incorporate the key points from this discussion into the final manuscript to enhance clarity. Thank you for positively reassessing our work!

---

### Author Response · Authors · 2024-11-19
**General Response**

We sincerely thank all the reviewers for their time and effort in evaluating our paper. We are pleased to see a shared recognition of its high quality. In particular, the reviewers noted that our paper is studying an important problem (GmGR), well-structured (Awai, Sd8N), clearly written (53Wx), easy to follow (Awai), easy to read (53Wx), and in good quality (Sd8N). We are also delighted that our idea and methodology received favorable feedback, being described as practical (GmGR, Awai), interesting (Awai), well motivated (Sd8N), impressive (53Wx, Sd8N), and sound (Sd8N). Additionally, we are pleased that the reviewers appreciated the extensiveness (53Wx) and comprehensiveness (53Wx, Sd8N) of our experiments, along with the theoretical aspects of our work (Awai, 53Wx, Sd8N).

We have addressed all of the reviewers' comments and, in accordance with ICLR policy, we have uploaded the revised PDF with changes from the initial version highlighted in blue for clarity. The key updates are as follows:

- (Table 3) We noticed that the pSGLD baseline results were accidentally omitted from Table 3, and we have now included them.
- (Figure 2) We have added the condition number plots to address the concerns raised by Reviewer 53Wx. These plots further clarify that EP enhances exploration without compromising convergence.
- (Appendix B.6 and B.7) All reviewers have raised concerns regarding computational overhead. In response, we have provided practical details on GPU memory allocation and wall-clock time during our experiments. Additionally, following constructive feedback from Reviewer Awai, we proposed representing the expanded matrices in a low-rank plus diagonal form to effectively reduce memory overhead. Furthermore, based on insightful comments from Reviewer 53Wx, we emphasized that introducing EP is more efficient than adding a burn-in period when considering actual wall-clock time.
- (Appendix B.8) Following Reviewer Awai's comment, we have conducted an additional comparison with the recent meta-learning approach by Kim et al. (2024) for improving SGHMC, using their official implementation to ensure reliable experimentation. By simply modifying the parameterization, we achieve competitive performance with their highly costly meta-learned sampler.
- (Appendix B.9) To clearly isolate the contribution of EP from the cyclical step size schedule, as pointed out by Reviewer GmGR, we present additional experimental results using a constant step size schedule. The additional results demonstrate that EP improves SGMCMC in a way that is orthogonal to the cyclical step size schedule.
- (Appendix E) We have attached a conceptual illustration as Figure 11 that provides an intuition why the parameter expansion induces the implicit step size scaling by comparing with the standard SGMCMC gradient update.

We will continue to make further revisions, taking into account any additional concerns raised by the reviewers. Once again, we sincerely appreciate the time and effort the reviewers have dedicated to our work!

---

### Author Response · Authors · 2024-11-25

Dear Reviewers,

Thank you again for your valuable feedback and thoughtful questions. We have provided responses addressing your concerns and shared an updated version of our manuscript.

As we approach the final days of the discussion period (with approximately two days remaining), we kindly encourage your participation in the discussion. Your additional comments and responses would greatly assist us in refining our work further.

We sincerely appreciate your time and effort in helping us improve our submission.

---

### Comment · Area_Chair_hA8d · 2024-11-25
**Discussions between reviewers and authors**

Time for discussions as author feedback is in. I encourage all the reviewers to reply. You should treat the paper that you're reviewing in the same way as you'd like your submission to be treated :)

---

### Meta-Review · Area_Chair_hA8d · 2024-12-20

**Metareview:**

This paper proposes PX-SGMCMC, where the main idea is to reparameterise the target random variable (in this work the neural network weights) as a product of several matrices and then conducting SGMCMC sampling in that augmented space ("parameter expansion"). Theoretical justification of the approach is provided. Experimentally the proposed sampler performed better in terms of improving sample diversity compared with a number of pre-conditioned SGMCMC methods, without incurring too much higher computational cost.

Reviewers overall commended on the paper's novelty and solid presentations, although they initially also raised questions regarding: the motivation of the approach, how the proposed approach compare with other pre-conditioning techniques (e.g., meta-learned ones), and the memory cost. During rebuttal the authors have improved paper with more explanations, and added experimental results regarding the requested comparisons and memory reduction.

After a brief read I think the paper is worth highlighting due to its novelty and solid development (in both theory and also extensive experiments). It should be of great interest for the Bayesian machine learning community.

**Additional Comments On Reviewer Discussion:**

Most of the major concerns were addressed during the reviewer-author discussion period.

---

### Decision · Program_Chairs · 2025-01-22

Accept (Poster)